# A biomimetic laminated strategy enabled strain-interference free and durable flexible thermistor electronics

Sanwei Hao [1], Qingjin Fu[1], Lei Meng[1], Feng Xu[1] & Jun Yang [1,2] ✉

The development of flexible thermistor epidermal electronics (FTEE) to satisfy high temperature resolution without strain induced signal distortion is of great significance but still challenging. Inspired by the nacre microstructure capable of restraining the stress concentration, we exemplify a versatile MXene-based thermistor elastomer sensor (TES) platform that significantly alleviates the strain interference by the biomimetic laminated strategy combining with the in-plane stress dissipation and nacre-mimetic hierarchical architecture, delivering competitive advantages of superior thermosensitivity ($-1.32\%$ °C$^{-1}$), outstanding temperature resolution (~0.3 °C), and unparalleled mechanical durability (20000 folding fatigue cycles), together with considerable improvement in strain-tolerant thermosensation over commercial thermocouple in exercise scenario. By a combination of theoretical model simulation, microstructure observation, and superposed signal detection, the authors further reveal the underlying temperature and strain signal decoupling mechanism that substantiate the generality and customizability of the nacre-mimetic strategy, possessing insightful significance of fabricating FTEE for static and dynamic temperature detection.

Precise and continuous epidermal temperature monitoring can facilitate the early prediction of cognitive states and abnormal physiological changes of the human body including cardiovascular diseases, pulmonary diagnostics, virus infection of COVID-19, and other syndromes[1–4]. Despite the thriving progress, the conventional rigid thermocouple conductors and infrared (IR) camera cannot achieve dynamic and continuous temperature detection in a body-comfortable manner and are susceptibly affected by the arbitrary movement of the target area. By contrast, the flexible thermistor epidermal electronics (FTEE), possessing the thermoelectric and wearable characteristics, has attracted immense interest as an alternative to overcome the drawback of rigid electronics by mechanically compliant to curved skin surfaces and receives a burgeoning amount of interest in areas ranging from artificial skin, biomimetic prosthetics, to health monitoring[5–8]. However, it is urgently desirable but still a formidable challenge to address the intrinsic compromise between

thermosensitivity and flexibility because the strain induced electrical signal interference and structure vulnerability inevitably lead to the signal distortion and performance deterioration under synchronous deformation together with skin.

To achieve mechanically agile and skin-integrable FTEE for stable and accurate temperature discrimination, many endeavors have been dedicated to exploiting the thermosensitive nanofillers including graphene[3], carbon materials (carbon nanotubes[9], carbon black[10]), metallic materials (Pt[1], Ag[11], and Au[12]), MXene nanosheets[13], and polyaniline[14] integrated composites (semiconductor) and stacking with dielectric layers to construct the intrinsically stretchable thermistor elastomer sensor (TES), encompassing the excellent thermal resolution and deformability along with obvious resistance signal to accommodate dynamic detection states. Despite the resistance-type signals can be identified easily and possess high signal intensity, the thermosensitivity of active layer and crack propagation would cause the similar resistance

[1]Beijing Key Laboratory of Lignocellulosic Chemistry, College of Materials Science and Technology, Beijing Forestry University, Beijing 100083, China. [2]State Key Laboratory of Pulp and Paper Engineering, South China University of Technology, Guangzhou 510640, China. ✉e-mail: yangjun11@bjfu.edu.cn

variations that act as side effects of signal fusion during accurate temperature monitoring. Additionally, there is a conceptual confusion that the merely independent temperature and strain signal acquisition mechanism is not equal to the interference-free temperature discrimination, because the structural variations from the initial equilibrium state during arbitrary deformation inevitably lead to the resistance changes to some extent, even though some temperature and strain bimodal sensors possess impressive thermal resolution in static state and show great potential in simulating skin response to multiple stimuli[9,14]. For example, Dong et al.[15] proposed a muscle-mimicking dual-sensory hydrogel-based sensor that relied on the polyaniline nanofibers (PANI NFs) as functional reinforcements to attain satisfactory thermosensation (1.6% °C$^{-1}$) and excellent gauge factor (GF) (-3.93 for strain range of 0–75%) with outstanding bimodal signal feedback and indication. Although there have been considerable approaches including multifunctional layer stacking[16,17], sensor array[2,18], and circuit off-setting[19] to achieve bimodal signal differentiation, the dilemma of signal distortion of these configurations is not well addressed since the intrinsic strain induced resistance still could be identified, yet also suffers from the complex and laborious fabrication steps, difficulties in analysis, structural vulnerability, mechanical mismatch between functional layers, and undesirable temperature resolution, impairing to a large extent of the sensing performance and hindering FTEE realistic application. Therefore, the ideal TES paradigm should provide exceptional thermosensitivity, mechanical stability, and avoids strain interference via a facile and universal method to reduce the strain sensitivity or restrain the deformation directly, enabling the accurate temperature monitoring of FTEE toward various scenarios.

Bioinspired structures, such as nacre-like structural hierarchy that correlated with heterogeneous and laminated arrangement, have attracted considerable attention but are still underused[20]. Many nacre-inspired approaches mainly focus on mimicking the hierarchical structure for mechanical robustness, while few efforts have taken interlocking and discontinuous characteristics of natural nacre into consideration[21,22]. Actually, gaining deep insight into the synergistic toughening effects of interlocking and rationally utilizing the heterogeneous arrangement would provide promising guidance for designing FTEE with the aim of achieving strain-tolerance and mechanical durability, but still lack theoretic demonstration and remain largely unexploited.

Herein, we propose a nacre-mimetic laminated strategy in combination with topological interlocking and alternating laminated architecture, enabling in-plane stress dissipation and deformation suppression for achieving high-fidelity temperature discrimination. The compliant, durable, and strain-interference free thermistor elastomer sensor (TES) is facilely fabricated by the layer-by-layer (LBL) assembly and compression assisted technique in tandem with the interlayer crosslinking agents of Fe(II) to stack the ultra-thin MXene nanosheets involved poly(vinyl alcohol)/TEMPO oxidated cellulose nanofibrils (TOCNF) composites as sensing layers, like the stacking up LEGO pieces. Remarkably, we verify that the dynamic interactions among MXene, TOCNF, and Fe(II) serve as interfacial crosslinking for alleviating interlayer stress concentration, and heterogenous adjacent layers act as charge barriers inhibiting electron migration, which collectively establish the internal resistance variation equilibrium to guide the performance optimization and satisfy the stringent requirement of the temperature resolution without signal distortion. The as-built TES that packaged by the thermal stable fluorinated ethylene propylene, demonstrates fascinating thermosensation (−1.32% °C$^{-1}$) without crosstalk, superior temperature resolution (0.3 °C), and ultrafast responsive time (7 s), along with excellent mechanical durability without undesired performance deterioration even subjected to extreme mechanical shock and over 20,000 folding fatigue cycles. More importantly, the unique strain insensitivity mechanism originates from interlocking laminated architecture rather than the special material compositions,

which shows great universality and can be extended to various nano-fillers including carbon black (CB), graphene (GO), carbon nanotube (CNT), and polyaniline (PANI). It is believed that this general and customizable nacre-mimetic strategy holds great promise and enriches a new perspective for facile fabrication of the FTEE, which may lead to prominent applications in healthcare management, smart prosthetics, and human-machine interface.

## Results
### Nacre mimetic designing
In the natural nacre, the hierarchically ordered structure of robust aragonite platelet featuring slippage mechanism offers the inherent stiffness[23,24], while the organic proteins promote the interlocking to restrain stress concentration for obstructing the excessive slippage or delamination of adjacent platelets (Fig. 1a). However, exploiting interfacial bridging for efficient platelet sliding mechanism and further modulating crack propagation or piezoresistive mediated resistance-type sensitivity is still rarely reported[25]. Fig. 1b, c schematically illustrate the proposed strategy for preparing TES with the nacre-like architecture and topological interlocking for structural stability and strain-interference free temperature sensing (Supplementary Fig. 1, detailed preparation process). Specifically, the MXene nanosheets are blended with PVA/TOCNF precursor solution (hereinafter referred as PTF) to prepare the PTF/MXene single layer via a typical blade-coating procedure (thickness = 1 mm). Prior to the following LBL assembly, the completely dried single layer (25 °C, ~48 h) was immersed into a high concentration of alkaline hydroxide solution (6 M, 20 min) to enhance the PVA crystallinity (Supplementary Fig. 2 and Note 1)[26], thereby constructed the dense polymer networks for mimicking the robust aragonite platelets in natural nacre (confirmed by XRD and DSC in Supplementary Figs. 3, 4). Thereafter, the multilayer composites were collected by stacking the single layer through a relatively straightforward and cost-effective LBL assembly associated with spraying FeCl$_2$ solution (1 M, 3 mL) between each adjacent layer to form interfacial bridging via the dynamic interactions among MXene, TOCNF, and Fe(II). With further assistance of compressing (20 MPa, 20 min) to promote the high alignment and remove the most of interlayer voids, the ultrathin and compact alternating laminated PTF/MXene/Fe composites (8 layers, 0.6 mm) were ultimately constructed as the sensing layer, then transferred and packaged into the thermal stable fluorinated ethylene propylene for thermistor elastomer sensor (TES). It is worth noting that the procedure variables including drying time, single-layer thickness, Fe(II) proportion, and compression time and strength, have been carefully screened to guarantee the optimal and reproducible performances of the samples (details in Methods Section).

Remarkably, the as-built flexible TES is superior to the commercial thermocouple since the robust hierarchical layer and dense interfacial bridging enable the mechanical toughening and strain insensitivity without sacrificing mechanical compliance, thus achieving the high-fidelity temperature acquisition under large deformations (Fig. 1d). In order to visualize the intrinsic disturbance tolerance versus deformation, a nonlinear geometric model is constructed and manifested by finite element simulation (Fig. 1e), where a predefined vertical stress field is applied as an exemplification to mimic the most frequently encountered force in daily life[21,27,28]. The result clearly verifies that the synergistic effects of robust interfacial bridging and compact heterogeneous arrangement lead to a heterogeneous stress distribution, where the stress is merely concentrated around in-plane adjacent bridging regions once the loading is applied, demonstrated as the main discrepancy on both exterior edge and inner edge. Upon further increasing the applied loading, the stress dispersion along the in-plane dynamic crosslinking among each layer is triggered to suppress deformation (Supplementary Movie 1). While this highly idealized model differs from the practical implementation, the exploration of the local stress

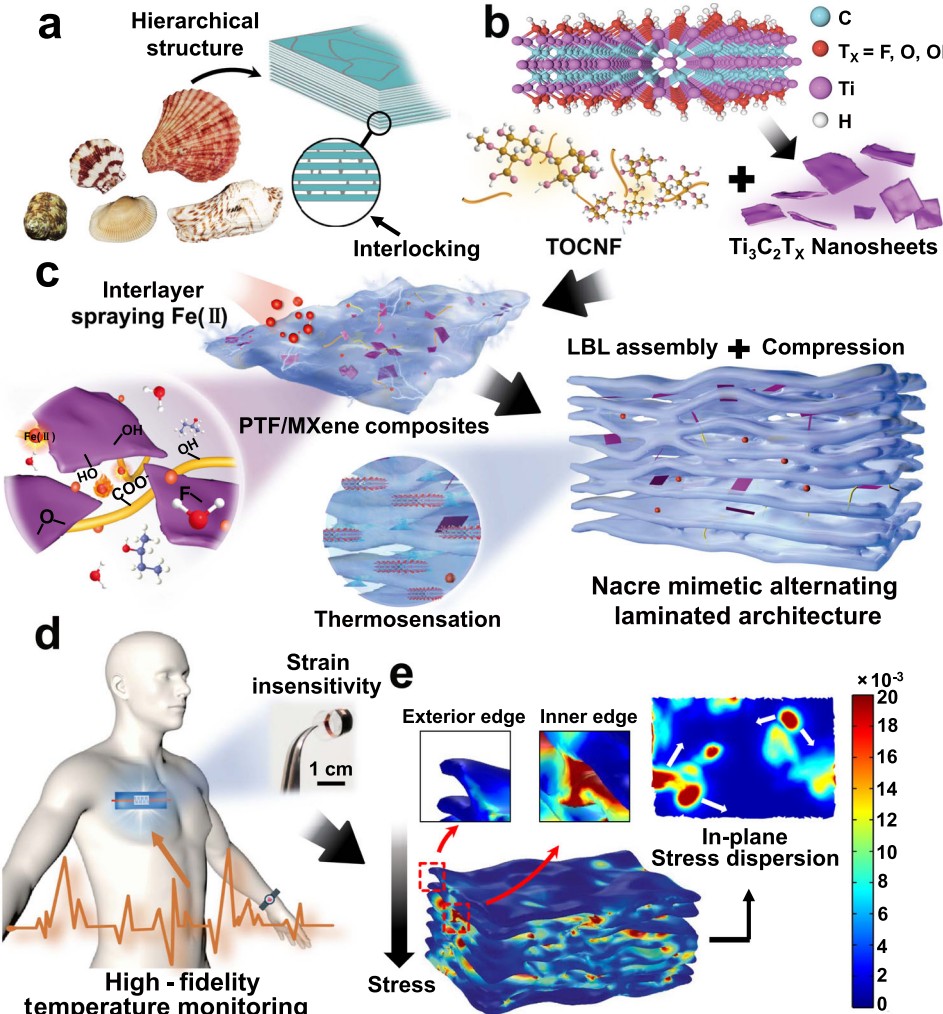

**Fig. 1 | Strain-interference free TES via a biomimetic laminated strategy.**
**a** Conceptional design inspired by the natural nacre. **b** Schematic overview for preparing sensing layer of TES with TOCNF and MXene building blocks in PVA polymer matrix, as well as spraying Fe(II) among adjacent layers for cross-linking.
**c** The proposed interfacial bridging between Fe(II) and functional groups (-OH, -COOH, and -F), as well as the facile and scalable LBL assembly and compression assisted technique. **d** Envisioned application of assembled TES for high-fidelity skin temperature monitoring. **e** Finite element simulation results of structural model demystify the interlayer stress concentration alleviation and in-plane stress diffusion.

evolution with deformation can vividly evidence the rationality of our strategy to tackle the issue of signal distortion by minimizing the deformation amplitude and inhibiting the conduction paths via interlayer barriers (further details discussed below).

## Interfacial bridging

From the perspective of effectively improving the thermosensitivity of TES, the prepared MXene nanosheets are selected as building blocks owing to their abundant terminal groups (F, O, and OH), large specific surface area, and high electrical conductivity[29,30]. The resultant MXene solutions demonstrate a uniform colloidal dispersion (Tyndall scattering effect) due to the electrostatic repulsion force among MXene nanosheets (Fig. 2a).The average thickness of 3.5 nm in atomic force microscopy (AFM) and the shift of the characteristic (002) peak (from 9.3 to 7.2°) in X-ray diffraction (XRD) patterns are in good agreement with the previous report[31], indicating the successful preparation of MXene nanosheets (Fig. 2b).

On the premise of MXene nanosheets guarantee the hypersensitive perception to temperature, the PVA elastomer materials with high crystalline and dense networks are regarded as supporting skeleton together with MXene nanosheets via hydrogen bonds to form elastic layer for mimicking the parallel "aragonite platelets" in the alternating laminated architecture. However, the relatively weak hydrogen bonds or Van der Waals force among the adjacent layers are insufficient for stable interfacial bridging[22]. To remedy this limitation, a straightforward and universal complexation is deliberately sought to act as robust interfacial bridging for coupling adjacent layers, such as the coordination bonds between carboxyl groups and metal ions[32–35]. In detail, the plant-originated TEMPO oxidated cellulose nanofibrils (TOCNF)[36,37], possessing numerous features involving uniformly microscopic size (0.8–2.0 um) (Supplementary Fig. 5), appealing mechanical properties, desirable modifiability, and the abundant functional groups (selective oxidation of C6 primary hydroxyl groups of cellulose to carboxyl groups), are supposed as the accessible reinforcements to interweave the MXene nanosheets and also act as the interlaminar crosslinking agent to interlock the adjacent layers via the coordination bonds with Fe(II). As illustrated by FTIR spectroscopy in Fig. 2c, the obvious peak shift from 1730 to 1720 cm$^{-1}$ and a new low intensity peak at 1633 cm$^{-1}$ substantiate the metal chelation between carboxyl groups of TOCNF and Fe(II). The digital photograph of the prepared PTF/MXene/Fe composites (8 × 12 cm$^2$) show an average thickness of ~0.6 mm with a mean error of 0.02 mm (Fig. 2d, e) and the surface morphology of adjacent layers is observed to examine the massive micro-protrusion distribution for interfacial bridging (Fig. 2f).

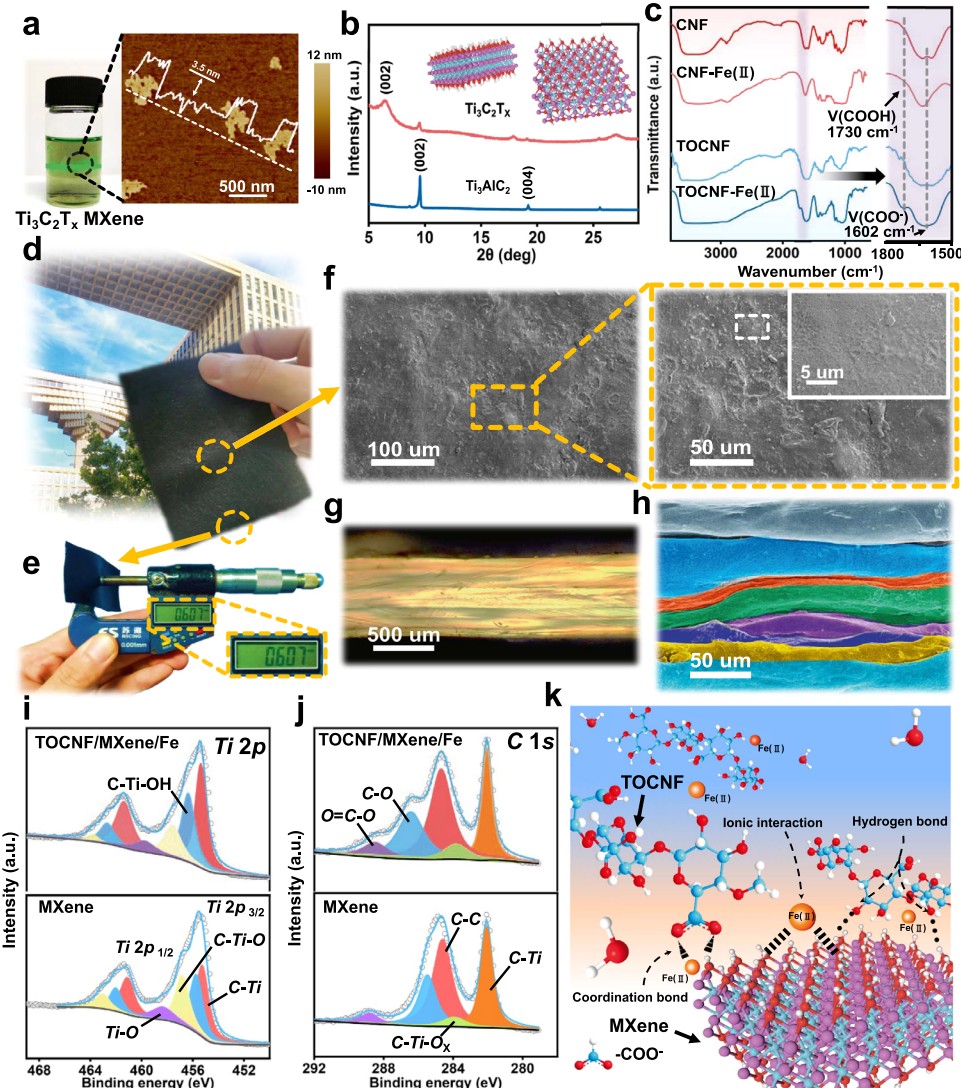

**Fig. 2 | Architecture and interfacial bridging characterization. a** The Tyndall scattering effect of $Ti_3C_2T_x$ (MXene) suspension and the corresponding delaminated MXene nanosheets by AFM image. **b** XRD patterns of $Ti_3AlC_2$ (MAX) precursor and MXene nanosheets. **c** FTIR spectra of CNF, CNF-Fe(II), TOCNF, and TOCNF-Fe(II), respectively. **d, e** Digital images of the scalable PTF/MXene/Fe composites with thickness of 0.6 mm measured by a micrometer (building in **d** designed by the School of Landscape Architecture of Beijing Forestry University). **f** SEM images of PTF/MXene/Fe composites with bulges and wrinkles on surface under different magnifications. **g** Cross-section polarizing microscopy and **h** colorized SEM images depict the compact alternating laminated architecture. **i** *Ti 2p* and **j** *C 1s* XPS spectra of MXene and TOCNF/MXene/Fe(II). **k** Illustration of interfacial bridging (coordination bonds, hydrogen bonds, and ionic interactions) between adjacent layers for topological interlocking.

Additionally, the cross-sectional polarizing microscopy and colorized SEM images validate that the adjacent layers are compactly integrated to form a dense lamellar microstructure, analogous to the hierarchically ordered structure of nacre (Fig. 2g, h). Previous studies proved that the cations including Fe(II), Mg(II), Co(II), and Ni(II) are likely to form weak alkoxide or coordinate bonds with the -OH, -F, and -O groups for immobilizing the MXene nanosheets[38], which is correlated with ions hydration energies (Gibbs free energies of various ions were listed in Supplementary Table 1). Although the trivalent ions with high valence possess large potential to bond with -OH groups, most of them, such as Fe(III), are also more likely to oxidize the MXene nanosheets[38]. For this concern, we thereby use Fe(II) rather than Fe(III) to bond with functional groups (-OH and -COOH) (Supplementary Fig. 6), and the mass ratio of $FeCl_2$ to MXene nanosheets is constant at 3:8 for preventing the excessive destruction of electrostatic repulsive force between the MXene nanosheets.

Additionally, X-ray photoelectron spectroscopy (XPS) provides more clues that the effective interfacial bridging is dependent on the synergistic interactions among TOCNF, MXene nanosheets, and Fe(II). The XPS spectrum of the TOCNF/MXene/Fe(II) confirmed the coexistence of C, O, Ti, and Fe elements that coincided well with the EDS mapping analysis (Supplementary Figs. 7, 8). In the high-resolution map of *F 1s* of MXene/Fe(II) profiles, the binding energy at 684.5 and 685.4 eV represents the bond of *C-Ti-F* and *Fe-F* (Supplementary Fig. 9), which is consistent with previous reports of the ionic interactions between MXene nanosheets and Fe(II)[38]. Notably, a slight binding energy of *C-Ti-F* shifted to 685.1 and 685.2 eV in TOCNF/MXene and TOCNF/MXene/Fe(II) *F 1s* profile, which is attributed to the hydrogen bonding between −OH of TOCNF and -F of MXene nanosheets. The *Al-F* peak could be derived from the residual Al atoms of the MAX precursor. Moreover, the *Ti 2p* and *C 1s* XPS profiles of MXene and MXene/TOCNF/Fe(II) are demonstrated in Fig. 2i, j where the *Ti 2p* spectrum can be fitted into 2p1/2 and 2p3/2 doublets because of the spin-orbit coupling effect[39]. The positive shift of *Ti-O* (458.3/459.9 eV) and *C−O* (285.5/286.4 eV) further suggest the multiple hydrogen bonds between MXene nanosheets and TOCNF. Overall, these results concordantly

corroborate the synergistic effects (coordination bonds, hydrogen bonds, and ionic interactions) among TOCNF, MXene nanosheets, and Fe(II) (Supplementary Movie 2), which collectively offer the strong interfacial bridging to couple the adjacent layers for topological interlocking as shown in Fig. 2k and may provide unambiguous guidance for designing interlayer coupling in various architectures.

Regarding the basic mechanical requirements of FTEE withstanding arbitrary deformations, the sensing layer as a pivotal component of the assembled TES should be mechanically resilient and robust, which weighs equal significance compared to strain-unperturbed thermosensation. For this concern, the important parameters configuration including the structural layer number (8) and the TOCNF contents (1 wt%) for the performance optimization in terms of skin-like Young's modulus ($0.35 \pm 0.032$ MPa) and a relatively high toughness ($50.62 \pm 1.53$ MJ m$^{-3}$) are demonstrated in Supplementary Fig. 10 and Note 2, which verify the efficient interfacial bridging and heterostructure achieving outstanding energy dissipation for synergistic toughening.

## Assembly of thermistor sensor for thermosensation

To satisfy accurate and real-time physiological temperature monitoring, a thermistor elastomer sensor (TES) is assembled where the PTF/MXene/Fe composites are connected with the copper wires to form ohmic interconnection and fully encapsulated by the commercially available and thermally stable fluorinated ethylene propylene (FEP) substrates (Fig. 3a). Notably, the thermal responsiveness is probed by the relative electrical resistances ($\Delta R/R_0$) versus temperature using water as heat transfer medium (Supplementary Fig. 11). Benefiting from an appropriate combination of both excellent compliance and ultrathin features, the assembled TES demonstrates desirable adaptability to skin and achieve the conformal skin attachment without obvious slippage or delamination throughout bending, twisting, and relaxing cycles (Fig. 3b). Strikingly, we investigate the $I-V$ relationship of TES at various temperatures (from 20 to 80 °C), which evidenced the significant decrease of resistance versus temperature, known as negative temperature coefficient (NTC) (Supplementary Fig. 12 and Movie 3). That is, the elevated temperature activates the electrons in the valence band and moves to the conduction band, resulting in more charge carriers for the thermoresistive effect. Considering the thermally activated charge carriers and the nacre-like hierarchical structure of the sensing layer of TES, the proposed thermosensitive mechanism is illustrated in Fig. 3c. On the one hand, the presence of thermally activated electrons provides rich charge carriers that contribute to the high hopping probability and the long-range hopping between MXene nanosheet junctions (Supplementary Fig. 13). On the other hand, the tight packing of MXene nanosheets during the LBL process facilitate the construction of thermally conductive pathways that reduce the energy barriers for electron hopping (Supplementary Fig. 14)[40–42]. To examine the crucial role of MXene nanosheets in thermosensation, the temperature coefficient of resistance (TCR) is quantified as Eq. (1):

$$TCR = [(R_T - R_0)/R_0)]/\delta T \tag{1}$$

where $R_T$ and $R_0$ represent the instantaneous resistance at measured temperature $T$ and reference temperature (20 °C), respectively. As expected, the TCR values dramatically increase from 1.04 to 1.37% °C$^{-1}$ in the large temperature interval ($\Delta T = 60$ °C, 20–80 °C) with the MXene nanosheets proportion from 1 to 10 mg ml$^{-1}$ (Fig. 3d), where TCR values far exceed the most of literature-reported thermistor epidermal sensors[10,13,18]. It should be mentioned that further doping exceeding 8 mg/ml yield a marginal improvement of TCR value about 0.05% °C$^{-1}$ because the notorious issue of stacking of adjacent MXene nanosheets (Fig. 3e), thus 8 mg mL$^{-1}$ (TCR, 1.32% °C$^{-1}$) is choose as an optimal content in the subsequent discussion. Another point worth

highlighting is that the TES sensors produced from different batches in the lab-scale setup possess high reproducibility (Supplementary Fig. 15a, b), and the thermal performances are hardly influenced by the ultrathin (80 um) and thermally stable FEP encapsulation layer (Supplementary Fig. 15c). Significantly, the $\Delta R/R_0$ versus temperature is fitted by nonlinear exponential line, and the $ln(R)$ versus $1000/T$ can be vigorously fitted by the Arrhenius model as Eq. (2):

$$In(R) = In(R_i) + B/T = In(R_i) + E_a/2k_B T \tag{2}$$

where $R_i$ is the resistance at an infinite temperature, $E_a$ is the thermal activation energy, $k_B$ is the Boltzmann constant, and the term $E_a/2k_B = B$ is the thermal index (Fig. 3f). In the temperature regime of 20–80 °C, $ln(R)$ versus $1000/T$ plot is fitted with high linearity ($R^2 = 0.97$), which emphasizes the thermal excited charge carriers to dominate Arrhenius-like temperature dependence and show the obvious NTC behavior. As exemplified in Fig. 3g, it can be seen that the resistance monotonically declined (from 7.11 to 2.98 kΩ) under stepwise temperature change from 32 to 80 °C that covers the entire temperature range required for human health monitoring, which further proves the NTC feature of TES.

More interestingly, the instantaneous temperature response of TES is investigated and found that the response time of TES shortens from 13 to 7 s in the case of elevating detection temperature from 22.4 to 70 °C (Supplementary Fig. 16). According to the real-time infrared (IR) images during temperature monitoring (Fig. 3h), the response time is roughly the same as the time of heat transfer equilibrium, indicating the fast and dramatic resistance variations are consistent with the great thermal convection under the high temperature condition. Besides, the detection stability and durability that reflect the ability to retain thermoelectric function and structural integrity are examined. As expect, the reproducibility of thermal response is recorded during cyclic heating and cooling tests (20–80 °C), where the heating and cooling curves almost overlap and the hysteresis is negligible (Fig. 3i). To better verify the stability and durability, the TES is imposed to different temperature cycles (low-, middle-, high-temperature ranges) and a long time interval (45 days) that afford extraordinary alternating temperature cycle life (40 cycles) without obvious signal fluctuation and outstanding TCR value retention (90.9%), putting emphasis on extraordinary durable operation ability, especially face with complicated temperature and long-term service conditions (Supplementary Figs. 17, 18). More promisingly, the subtle temperature variations (0.3, 1, 2, and 5 °C) can be accurately and reproducibly discriminated (Fig. 3j and Supplementary Fig. 19), suggesting that the TES possesses unprecedentedly high temperature resolution for detecting tiny temperature signals and guarantees the most rigorous tests for daily precise temperature monitoring.

As a quick summary, all the discussions above confirm the overwhelmingly collective performances of TES including exceptional thermosensation (1.32% °C$^{-1}$), wide operating temperature range (20–80 °C), rapid response time (7 s), alternating temperature cyclic stability (40 cycles), and desirable temperature resolution (0.3 °C) based on the thermosensitive MXene nanosheets and compact alternating laminated architecture, manifesting its ability to accurately and repeatedly monitor temperature under complex conditions. As comparatively described in Fig. 3k and Supplementary Table 2, the salient merits of thermosensitivity and wide operating temperature range of TES, together with structure strategy induced strain insensitivity(discussed hereafter), are superior to the state-of-the-art counterparts that relied on various thermal nanofillers (carbon, graphene, PANI, Pt, Ag nanowires, etc.), highlighting the competitiveness of as-built TES and portraying a bright prospect in applications of the FTEE[2,3,10,11,13–16,18,19,41–58].

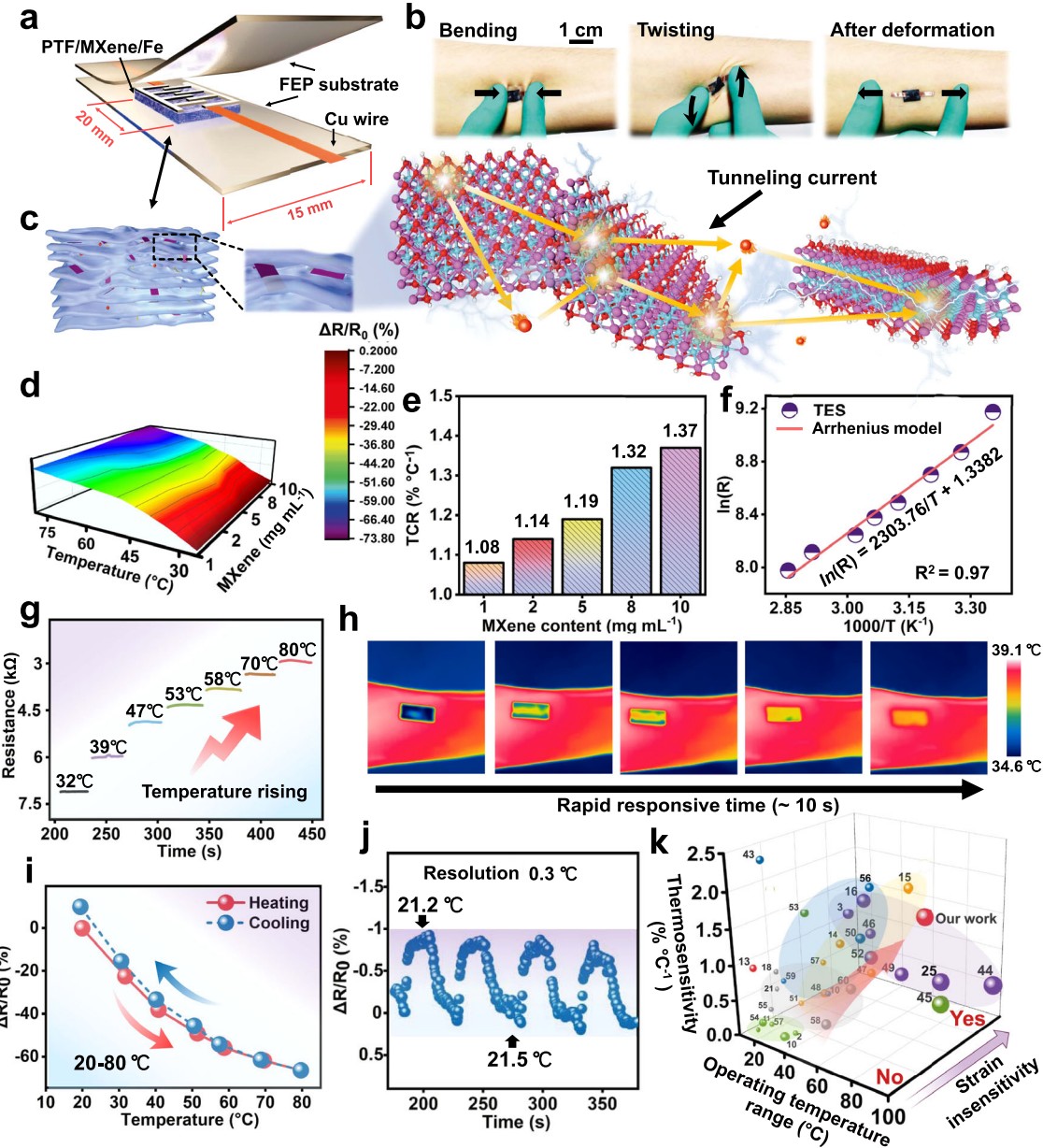

**Fig. 3 | Comprehensive thermosensitive performances. a** Illustration of TES composed of alternating laminated PTF/MXene/Fe composites, FEP encapsulation, and Cu wires, and **b** the seamless contact on the forearm along the curved contours without slippage/delamination throughout bending, twisting, and relaxing cycles. **c** The thermosensation mechanism rely on the thermally activated tunneling current passed through MXene nanosheet junctions. **d** The relative resistance variations and **e** the corresponding temperature coefficient of resistance (TCR) of TES with different MXene nanosheet concentrations (1–10 mg/mL) over a wide temperature range (20–80 °C). **f** Arrhenius plot depicting the linear dependence of $ln(R)$ versus temperature (1000/$T$). **g** The negative resistance response to gradient temperature. **h** The infrared (IR) images of thermal energy transfer during the skin temperature detection. **i** The negligible temperature hysteresis in cyclic heating and cooling cycles (20–80 °C). **j** The stable and repeatable resistance variations with high temperature resolution (0.3 °C). **k** Performance comparison in terms of thermosensitivity, operating temperature range, and the strain insensitivity between the assembled TES and various nanofiller-based FTEE reported in the literatures (including carbon, graphene, PANI, Pt, and Ag nanowires, and the detailed in Supplementary Table 2).

## Structure stability and strain insensitivity

The aforementioned investigation concretely validates the accurate and reliable thermosensation via the thermistor variation of as-built TES in the static condition. Promisingly, the TES also exhibits strain-interference free and stable thermosensation performances under dynamic scenario owing to the unique nacre-mimetic architecture and efficient interfacial bridging of sensing layer. The Fig. 4a further deciphers the structural organization: (1) The TOCNF interweaves MXene nanosheets as 1D building blocks to form the thermosensitive network (Supplementary Fig. 20, FTIR spectrum and XRD analyses)[37]. (2) The dense PVA polymer chains serve as 2D framework and offer abundant

active sites for mimicking the aragonite platelets in natural nacre (Supplementary Figs. 21, 22). (3) The dynamic interactions (coordination bonds, hydrogen bonds, ionic interactions) among MXene, TOCNF, and Fe(II) dominate the interlocking of adjacent layers that construct the compact 3D nacre-mimetic architecture. As revealed in confocal laser scanning microscope (CLSM) images (Fig. 4b), the surface morphology of pristine PVA is relatively smooth with few protrusions, while the bulges and wrinkles become apparently on the surface of PTF, PTF/MXene, and PTF/MXene/Fe composites along with the increased roughness (Sa, average roughness) from 4.36 to 11.21. These relative rough microstructures are expected to allow sufficient

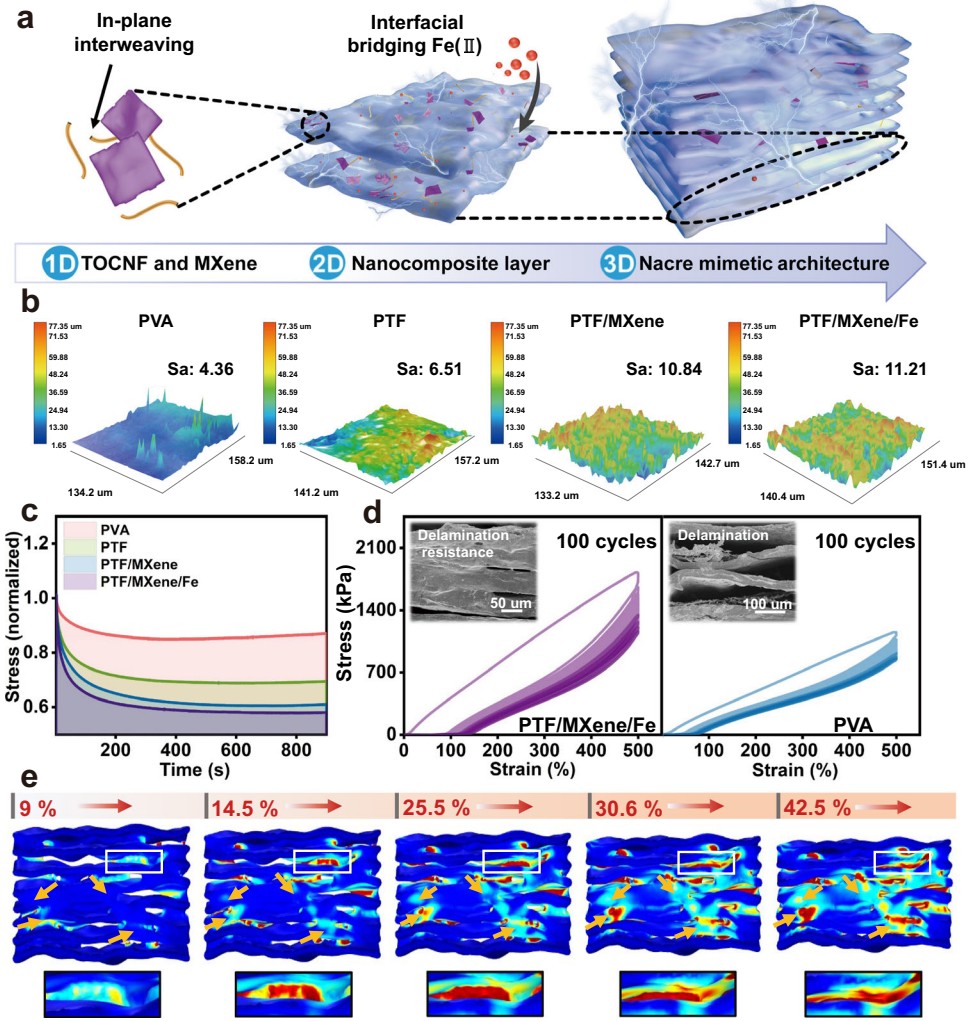

**Fig. 4 | Alternating laminated architecture and efficient interlocking enable strain insensitivity. a** Compositions from 1D to 3D for the nacre-mimetic architecture of sensing layer. **b** Confocal laser scanning microscope (CLSM) images show the surface microstructure for interfacial bridging, and **c** the stress relaxation curves stretch at 100% strain for PVA, PTF, PTF/MXene, and PTF/MXene/Fe composites, respectively. **d** Cyclic loading–unloading tensile curves (100 cycles, 500% strain) for PTF/MXene/Fe and PVA composites, the inset showing the cross-section SEM images after 100 stretching cycles. **e** Finite element simulation of stress distribution within alternating laminated architecture under the vertical uniaxial press of different deformation ratios (9, 14.5, 25.5, 30.6, and 42.5%, respectively).

contact area to expose hydrophilic groups for establishing strong interfacial bonding on the lamellar polymer layers, which is also corroborated by the surface hydrophilicity changes (Supplementary Fig. 23). Indeed, the stress relaxation test reveals that the PTF/MXene/Fe composites possess the maximal release of applied force at the constant strain of 100% and far exceed that of the PVA, PTF, PTF/MXene counterparts (Fig. 4c). Likewise, a pronounced hysteresis loop with the larger residual strain is noted from the successive loading–unloading curves in comparison with that of PVA, resulting in exceptional recovery efficiency (~73.57%) even after 100 fatigue cycles at strain of 500% (Fig. 4d). The inset cross-section SEM images further clarify the robust interface of PTF/MXene/Fe composites in contrast to the delamination and slippage between adjacent layers of PVA that are negatively characterized by the appearance of extra interlayer voids, inferring interface failure and weak fatigue resistance owing to the weak interlayer van der Waals interactions and hydrogen bonds. This structural evolution is in accordance with the above results to confirm that the multiple dynamic interactions and coordination bonds acting as "sacrificial bonds" tend to preferentially dissociate for promoting the stress dispersion aiming at temperature monitoring without strain interference.

To better elucidate the underlying mechanisms of both heterogeneous laminated structure characteristics and tough interface for strain insensitivity, the finite element simulation is adopted to quantitatively analyze the lamellar domains and sheet orientation in alternating laminated architecture and track their evolution as a function of vertical stress field (Fig. 4e, Supplementary Fig. 24a–d and Note 3). The varying degrees of sensing layer deformation (9, 14.5, 23.5, 30.6, and 42.5%) are analyzed. It can intuitively see that the extensive stress distribution (bright color regions) underneath the topological interlocking and the numerous in-plane stress diffusion (noted as arrows) following tortuous paths (dynamic cross-linking) are distinctly intensified throughout enlarged contact regions of adjacent layers, in line with following speculations: (1) Accompanying with external loading, the interfacial bondings that mainly consist of the coordination and hydrogen bonds facilitate stress distribution to in-plane cross-linking. (2) When the maximum principal stress of the contact units reaches the critical intensity, the bending and buckling of topological interlocking that located on the disordered central parts dominate the structural deformation, and then the in-plane stress diffusion achieve around the main contact element of adjacent layers to prevent the interfacial slippage and

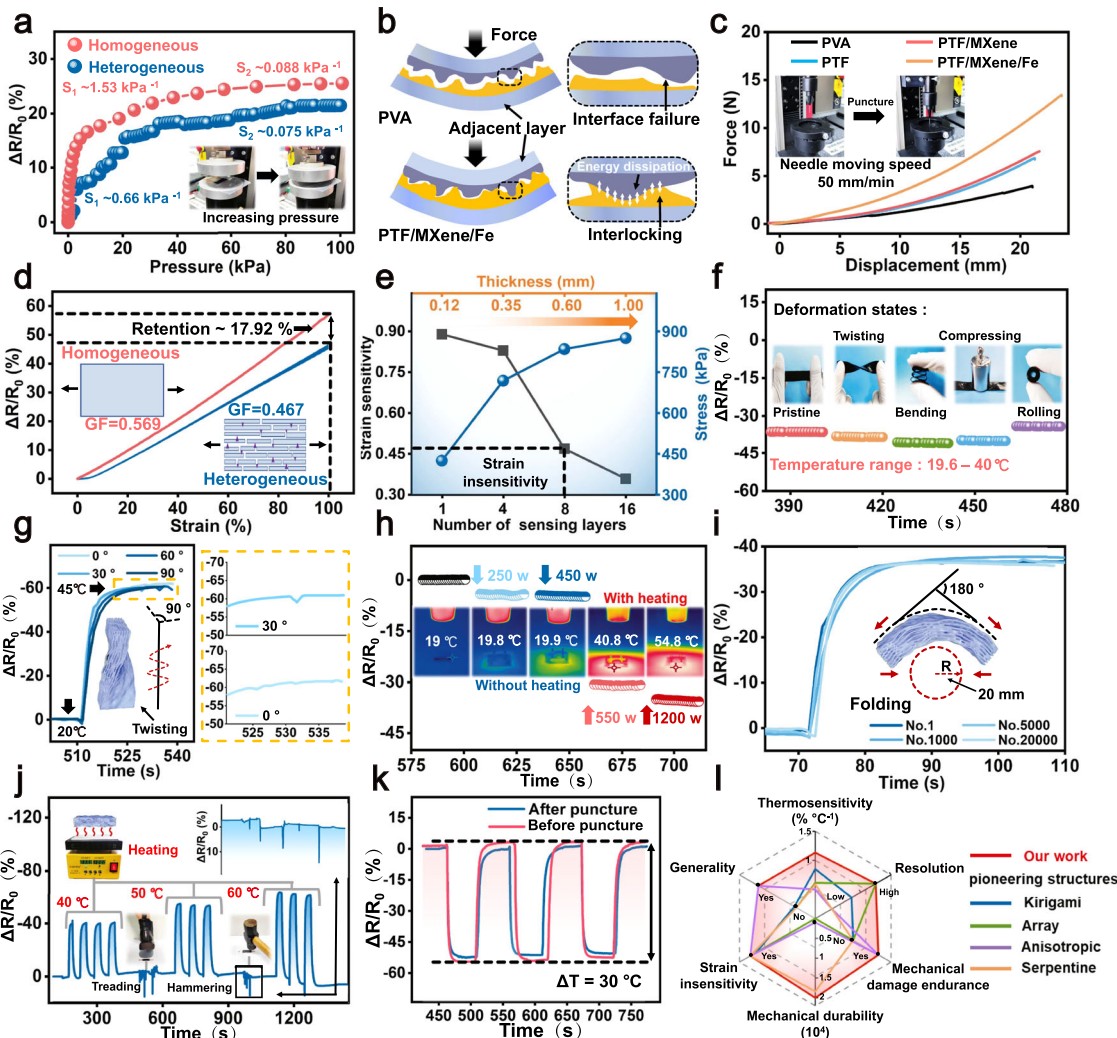

**Fig. 5 | Strain-unperturbed and stable thermosensation. a** The weaken piezo-resistive effect in the applied pressure region (0–100 kPa) for TES with interlocking laminated architecture. **b** Schematic diagram of the topological interlocking for eliminating the interface failure. **c** Puncture resistance test (needle diameter = 900 um, loading speed = 50 mm min⁻¹) of PVA, PTF, PTF/MXene, and PTF/MXene/Fe composites, respectively. **d** Unobvious gauge factor (GF) in 100% strain range for TES with interlocking laminated architecture in contrast to the TES with homogeneous structure. **e** The correlation of sensing layer number of TES with strain sensitivity and stress at 100% deformation, along with the thickness increasing. **f** The stable temperature detection ($\Delta T$ - 20 °C) free of strain interference under various harsh deformation states (twisting, bending, compressing, and rolling). **g** The high-fidelity temperature discrimination of TES under different initial torsion angles (0, 30, 60, and 90 °) to simulate the contact with curved skins. **h** The undistorted thermosensation against the hairdryer air flow (19.8, 19.9, 40.8, and 54.8 °C) under different output powers (cold wind 250 and 450 W, hot wind 550 and 1200 W). **i** Stable temperature signal waveform ($\Delta T$ - 20 °C) even after 20000 folding fatigue cycles, the inset of a schematic depiction of folding state (radius - 20 mm, angle - 180°). **j** The low resistivity pulses of temperature monitoring (40, 50, and 60 °C) upon extreme mechanical impacts (treading by a volunteer and impacting via a hammer). **k** Thermosensitive performance retention even after heavy mechanical puncture (pin diameter = 200 um). **l** Radar plots comparing between our design and engineering structural strategies in literatures for fabricating interference-free flexible thermistor epidermal sensors (details in Supplementary Table 3).

crack propagation, thus reflecting the strong interfacial interactions with various surfaces and mechanical interlocking coupled behaviors for deformation suppression (interfacial interactions and strain-tolerant thermosensation described in Supplementary Figs. 24e, 25). Since the strain sensitivity is strongly dependent on the formation and destruction of conductive pathways that correlate with the structural variations, while the efficient interfacial bridging with in-plane stress dispersion mechanism restrains the deformation and the heterogeneous architecture with adjacent layer barrier minimizes the charges transfer. According to the comparison of the TES possessing homogeneous elastomer matrix, we speculate that the synergistic effects of interlocking and nacre-mimetic architecture of TES collectively facilitate the strain-unperturbed thermosensation (revealed in Supplementary Fig. 26).

## Sensing performance

To quantitatively decipher the strain insensitivity of TES, both pressure sensitivity (S) and gauge factor (GF) are calculated, which defined as the ratio of relative resistance changes ($\Delta R/R_0$) to the applied pressure or strain, respectively. A type of reference sensor, defined as TES with homogeneous structure, only in absence of layer-by-layer assembly and interlayer spraying processes is fabricated for comparison. Figure 5a depicts the $\Delta R/R_0$ as a function of the applied pressure levels, and TES with homogeneous structure shows a two-stage nearly linear response (1.53 kPa⁻¹ of 0–12 kPa and 0.088 kPa⁻¹ of 12–100 kPa), while TES with interlocking laminated architecture presents an extremely weaken piezoresistive effect (0.66 and 0.075 kPa⁻¹ in both 0–20 kPa and 20–100 kPa). The pressure sensitivity is reduced about 2.3 times and the sensing range is extended to 0–20 kPa in the first

piezoresistive response interval. Practically, the compact heterogeneous architecture of sensing layer (PTF/MXene/Fe) is coupled by effective interfacial bridging, either already under physical contact or only marginally separate to hinder the creation of new conduction paths. Furthermore, the strong interfacial bridging alleviates the stress concentration for in-plane stress dissipation (Fig. 5b), giving rise to the deformation suppression and reveals as limited $\Delta R/R_0$. As a further proof, the puncture resistance and tearing tests of PTF/MXene/Fe composites are conducted, and the puncture force (13.42 N) and fracture energy (591 kJ m$^{-2}$) are over 1.76 and 1.30 times higher than that of PVA counterparts, respectively (Fig. 5c and Supplementary Figs. 27, 28). These striking out-plane loading tolerance further reflects the coordination bonds as interlocking for preventing stress concentration. Simultaneously, the energy dispersive spectrometer (EDS) mapping in Supplementary Fig. 29 provides direct clues that the Fe element as the interfacial bridging agent predominantly exists in the adjacent layers, confirming our assumption that coordination bonds play an indispensable role in facilitating the stress dispersion for strain insensitivity. Besides, Fig. 5d illustrates the decreased resistance variation about 17.92% of TES over the entire skin stretching range (0–100%), together with limited detection sensitivity (GF ~ 0.467) that far below the level of TES (GF ~ 0.569) with homogeneous structure and most previous flexible electronics[59,60]. From a more quantitative analysis, the strain-unperturbed performances can be optimized by tailoring the structural parameters like the layer number (Fig. 5e and Supplementary Fig. 30). Typically, the TES shows diminishing strain sensitivity from 0.89 to 0.36 with the increasing of the layer number (1–16, thickness from 0.12 to 0.99 mm), while the tensile stress at 100% deformation increases from 426 to 876 kPa. These results are consistent with the finite element simulation (Fig. 4e), meaning that the robust multiple layers as building blocks not only act as the electron transfer barrier for inhibiting conductive paths formation, but also contribute to the synergistic toughening in conjunction with interfacial bridging for restraining structural deformation, thereby realizing the thermosensation without strain-mediated signal distortion.

To be more intuitive, we extend the thermomechanical decoupling of TES to various large deformation states (twisting, bending, compressing, and rolling) with a constant temperature differential ($\Delta T$) of 20 °C, which shows a negligible effect on the thermosensation (Fig. 5f). We also inspect the relative resistance variations ($\Delta R/R_0$) versus temperature (20–45 °C) associated with different torsional angles (0, 30, 60, and 90°) of TES to simulate temperature monitor on curvilinear and dynamic skin surface (Fig. 5g), and the twisting induced signal variations are also largely suppressed (30°, $\Delta R/R_0$ ~ 0.84), which even remain below the maximum temperature resolution ratio (0.3 °C, $\Delta R/R_0$ ~ 1), verifying the validity of feeding back the subtle skin temperature changes without signal distortion. Remarkably, the high-accuracy temperature discrimination of TES is also manifested via monitoring hairdryer temperature under different motor powers (Fig. 5h and Supplementary Fig. 31), where the lower power state (cold wind, ~250 and 450 W) without temperature variation could not be discriminated (merely the wind pressure induced deformation). Conversely, the higher power state (resistance wire heating for hot wind, ~550 and 1200 W) generates a hot flow (40.8 and 54.8 °C) and leads to distinct and stable (up to 70 cycles) relative resistance variations under high temperature conditions. These results agree well with the computational predictions presented earlier (Fig. 4e), providing solid evidence of the strain insensitivity TES for accurate temperature detection. Considering the inevitable and high frequency mechanical manipulations in the wearable experiences, an exceedingly favorable feature, yet rarely reported, is the structural stability upon deformation to maintain the long-term temperature monitoring without signal distortion or parameter recalibration[47–52]. For mimicking the large and continuous deformation, the cyclic folding fatigue tests (20,000 cycles) are examined with a constant folding radius ($R = 20$ mm) on

sensing layer and the folding angle is changed from 0 to 180° (Fig. 5i). Promisingly, the output $\Delta R/R_0$ curves for temperature monitoring (20–37 °C) undergoing repeatable folding cycles are well maintained without obvious signal attenuation ($\Delta R/R_0$ variations, <0.7%) and the sensing layer exhibits no delamination, reflecting the strong interfacial bridging contributed to structural stability, thermosensation reliability, and long-term durability against large and repeated deformations. Additionally, it is urgent to examine mechanical impact resistance and damage tolerance because the application of TES may encounter drastic destruction or other unexpected situations such as the user sitting or lying down. The continuous temperature acquisition (40, 50, and 60 °C) upon undesirable mechanical shock is evaluated via consecutive treading (10–20 kPa) and hammering (5–15 kPa) (Fig. 5j and Supplementary Movie 4). Specifically, the high loading force induced resistivity pulses exhibit distinct features in terms of amplitude and signal duration where the variable amplitude and sharp pulses ($\Delta R/R_0$, ~3–13) could be easily distinguished with obvious and smooth thermoresistance waveforms ($\Delta R/R_0$, ~38–59). Alternatively, Fig. 5k unveils that the temperature response profiles between normal and punctured TES almost overlapped ($\Delta R/R_0$ ~ 50, $\Delta T$ ~ 30 °C) along with synchronous waveform peaks with negligible hysteresis. The above results imply the superior signal stability and interference-free temperature detection upon large deformation and extreme mechanical damages of TES in contrast to structural vulnerability of microcircuit printing and electronic patch designs[52,61]. Overall, the intriguing strain insensitivity and impact resistance features are closely correlated with the weak adjacent layer conduction pathway and efficient stress diffusion as derived from the heterogeneous structure and interfacial bridging, bringing fruitful inspiration to fabricate the mechanical robust and accurate FTEE.

More promisingly, the universal strain insensitivity originates from the unique alternating laminated architecture rather than the deliberate material choice, and various building blocks (e.g. CB, GO, CNT, and PANI) can be employed to prepare the TES, which also presents the accurate temperature acquisition without strain interference and achieves the tailorable TCR values from 1.19 to 1.50% °C$^{-1}$ (Supplementary Fig. 32), confirming the universality and customizability of the biomimetic laminated strategy for solving the strain interference issues and designing the thermosensitive materials with rational functionality. In the context of substantially improving the comprehensive performance of the TES, the proposed strategy becomes more competitive to the reported structural counterparts in several key metrics including thermosensitivity, resolution, mechanical damage tolerance, durability, strain insensitivity, and generality (Fig. 5l and Supplementary Table 3)[6,12,42,43,45,46,62–64], we foresee that this biomimetic laminated structural paradigm holds great promise and represents a credible new approach for the development of accurate and flexible thermistor electronics.

### Proof-of-concept of high-fidelity temperature discrimination

The establishment of a temperature perception system via TES is expected to achieve the early diagnosis of health status as conceptual representation in Fig. 6a. With regard to further confirming the application potential, several competitive advantages including impressive temperature resolution, long-term lifespan, fast response time, and precise temperature detection are given priority of consideration with the comparison of commercial T-type thermocouples (diameter = 0.3 mm, resolution = 0.2% +1 °C). As a demonstration, physiological temperature monitoring of different body parts (mouth, forehead, armpit, chest, and wrist) is conducted by conformal attachment of the assembled TES on the skin (Fig. 6b). Specifically, the surface temperature distribution is accurately detected through the relative resistance profiles and visualized by an IR camera (mouth ~35.5 °C, forehead ~34.7 °C, armpit ~36.5 °C, chest ~32.3 °C, and wrist ~30.7 °C), where the pulse waveforms are highly consistent with

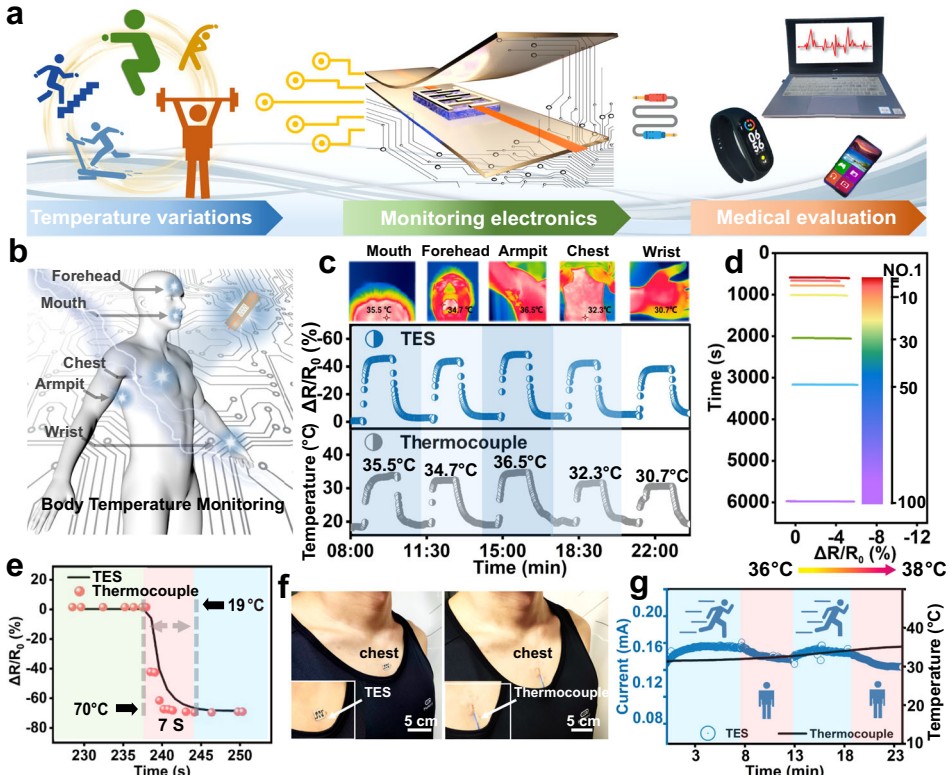

**Fig. 6 | Applicability assessment of TES in both static and dynamic scenarios.**
**a** Conceptual demonstration of thermistor electronic integration for the dynamic temperature monitoring. **b** Schematic illustration of temperature detection upon various positions of body (mouth, forehead, armpit, chest, and wrist), and **c** the comparison of TES (middle panel) with thermocouple (bottom panel), the IR images reflecting real-time temperature (upper panel). **d** Consecutive temperature identification for 100 cycles from 36–38 °C to mimic the continuous "fever indication". **e** The comparable temperature response time of TES and thermocouple upon large temperature variations (19–70 °C). **f** Optical images of the temperature monitoring by conformal contacting the TES (left panel) and thermocouple (right panel), and **g** the real-time current variations of TES without strain induced signal distortion during both the volunteer running (speed = 10 km h⁻¹) and rest sessions in contrast to the imprecise skin temperature monitoring of thermocouple.

that of measured by the rigid thermocouple thermometers that suffer from the drawback of subsiding the wearing comfort (Fig. 6c and Supplementary Fig. 33). In addition, the stable and repeatable temperature acquisition is achieved in the consecutive temperature cyclic situation (100 cycles, 36–38 °C) along with a negligible signal deviation ~5.5%, indicating the durability of repeatable temperature response that possess profound importance of TES for the early prediction of abnormal physiological changes (e.g., fever, infection, and heat stroke) in daily life, especially in the case of the thermocouple and IR camera cannot keep up with continuous monitoring demand (Fig. 6d). It is worth noting that the long-term skin contact (over 24 h) does not cause obvious skin redness or allergies, this phenomenon may be ascribed to the commercial FEP encapsulation possessing excellent skin-friendly capability and nontoxicity (Supplementary Fig. 34).

More interestingly, the fast response time ($\Delta T$ ~ 50 °C, 7 s) on par with that of the thermocouple is achieved (Fig. 6e), highlighting the potential of instantaneous temperature detection to satisfy the great demand of healthcare management. Additionally, the TES and thermocouple are conformably mounted to the volunteer's chest by using medical tape as a typical dynamic scenario, and the instantaneous temperature is recorded and validated through the IR camera (Fig. 6f and Supplementary Fig. 35). One should note that the $\Delta R/R_0$ is not normalized by current curves in order to intuitively demonstrate the undistorted signals, regardless of the strain interference caused by skin wrinkling or fabric friction. Indeed, the amplitude and duration of the current signals are highly consistent with the body temperature changes in the exercise time interval (Fig. 6g), where the downward trend of curve occurs owing to the heat loss after cessation of running

(0.195–0.171 mA, $\Delta T$ ~ 1.2 °C from 8 to 12 min) and the current goes up again until the volunteer running again (0.176 to 0.191 mA, $\Delta T$ ~ 1.2 °C from 13 to 18 min). Conversely, the thermocouple struggles with dynamic temperature acquisition caused by body movement which exclusively deems the intrinsic stiffness that hard to achieve the compliant skin contact, giving rise to imprecise signal identification. Lastly, the collective performances of the TES also inspire further exploration for environmental recognition that stems from its high temperature dependency (temperature is a typical daily and seasonal variable). As a proof-of-concept, a temperature monitoring system is built where the TES can automatically respond temperature-mediated resistance signals and lit a lamp within the 220 V alternating current system at night (Supplementary Fig. 36 and Movie 5). The application of our TES operating in practical environments demonstrates its promising potential for next-generation health-care monitoring and intelligent human-machine interactions.

## Discussion

Compared with the previous thermistor electronics in literature[64,65], the as-built TES overcomes the mechanical compliance limitation of conventional rigid thermocouple and IR camera, which also demonstrates the merits of compactly heterogeneous adjacent layers for minimizing external impacts and regulating the crack propagation or piezoresistive mediated resistance mechanism. Therefore, the assembled TES via general and customizable nacre-mimetic strategy successfully accomplishes accurate temperature monitoring without strain interference, showing harbor tremendous practical application value and potential in personal healthcare diagnosis. Despite the

proposed strategy has adequately addressed the typical issues of FTEE, the future research urgently deserves more efforts. From a design perspective, the enhancement of interfacial integration and the optimal dielectric layer configuration for essentially stretchable electronics should underway to promote the possibility of interdisciplinary design. From an assembly perspective, more efforts should focus on wireless acquisition and transmission to improve the level of miniaturization and integration that represent the next major steps towards realistic application.

In summary, we propose a promising nacre-mimetic laminated strategy to address the long-standing strain interference in thermistor electronic applications from structural aspects (summarized in Supplementary Fig. 37). This work reflects remarkable innovation in both structure design and performances: (1) We emphasize the efficient interface coupling, in-plane stress diffusion, and crack propagation suppression in the proposed biomimetic assemblies for strain insensitivity, thus overcoming the compromising between deformability and thermosensation. (2) The prepared TES features transformative performance improvement of superior thermosensitivity (1.32% °C$^{-1}$) and extraordinary resolution ratio (0.3 °C), together with unprecedentedly sensing durability capable of temperature acquisition after 20,000 folding fatigue cycles and even subjected to harsh mechanical shock. (3) The substantial progress of unparalleled temperature discrimination without signal distortion is achieved in the exercise scenario. It can be anticipated that the proposed customizable nacre-mimetic architecture, scalable preparation method, simple components configuration, and general interfacial bridging strategy will open up a facile and economic yet effective methodology for high-fidelity thermistor epidermal electronics and show broad application prospects in the fields of healthcare electronics, artificial prostheses, and intelligent robots.

## Methods

### Materials
The 400 mesh MAX phase (Ti$_3$AlC$_2$), poly(vinyl alcohol) (PVA, $M_w$ = 205,000 g mol$^{-1}$), NaOH, ferric chloride tetrahydrate (FeCl$_2$.4H$_2$O), lithium fluoride (LiF), and hydrochloric acid (HCl) were purchased from Beijing Chemical Reagents Company. The TEMPO-mediated oxidized cellulose nanofibril suspensions (TOCNF, -5.0 wt%) were purchased from Tianjin University of Science and Technology. Graphene (GO), carbon nano tube (CNT), polyaniline (PANI), and carbon black (CB) were obtained from Maclean Reagents Co. LTD. All chemicals are analytical grade and used without further purification.

### Fabrication of delaminated MXenes
Delaminated Ti$_3$C$_2$T$_x$ (MXene) was fabricated via the well-defined LiF/HCl method[6]. Initially, LiF (2 g) was added to 9 M HCl (40 mL) with magnetic stirring for 10 min to gain the uniform solution. Subsequently, 2 g Ti$_3$AlC$_2$ powder was slowly added to the above solution and continuously stirred at 35 °C, 300 rpm for 24 h. Then, the etched samples were centrifuged in deionized water at 5000 rpm (10,000 g) for 10 min until pHå 6 to obtain the dark green supernatants. After ultrasonic treatment for 1 h under nitrogen atmosphere, the multilayered MXene nanosheets solution was centrifuged at 3500 rpm for 30 min. Finally, the collected upper suspension was further freeze-dried to obtain MXene nanosheets powder and stored at 4 °C before usage.

### Preparation of sensing layer with alternating laminated architecture
Typically, the PVA powder (80 mg mL$^{-1}$) and TOCNF suspensions (5 wt%) were dissolving in deionized water at 90 °C by magnetic stirring (350 rpm, 12 h) to gain a uniform mixture (60 g). After cooling, the MXene nanosheets (8 mg mL$^{-1}$) were added to the above mixture by the assistance of sonication (30 min) under nitrogen environment. Subsequently, the PTF/MXene/Fe composites were collected through

a directional blade coating machine (100 mm min$^{-1}$) and dried completely (25 °C, 48 h). The resultant specimens were then immersed in high-concentration alkaline hydroxide (6 M, 1L) for 20 min and washed with water to permanently stabilize the crystalline domain of the single layered composites. After spraying with Fe(II) solution (1 M, 3 mL) between each layer, the stacked composites encapsulated by fluorinated ethylene propylene (FEP) were pressed via thermocompressor (20 MPa, 20 min) to fabricate the PTF/MXene/Fe composites with alternating laminated architecture for TES.

### Characterizations
Scanning electron microscopy (SEM) observation and energy dispersive spectroscopy (EDS) were performed on field emission scanning electron microscope (SU8010, Hitachi). The morphological observation was carried out using a polarizing light microscope (BX43, Olympus). The transmission electron microscopy (TEM) images were obtained at an accelerating voltage of 80 kV (JEM-1010, JEOL). The Atomic Force Microscope (AFM) images were obtained on Bruker Multimode 8, Germany. The average surface roughness and 3D profilometry of the composites were measured via true color confocal microscopy (CSM700, Zeiss). XRD patterns were recorded on the Bruker D8 Advance Diffractometer equipped with Cu Ka radiation ($\lambda$ = l.5406 Å). The interfacial cross-linking was examined by using an X-ray photoelectron spectroscopy (XPS, ESCALAB 250XI, Thermo Scientific) with Al K$\alpha$ radiation. The coordination bonds were verified by using a UV–visible spectrometer (TU-1901, Puxi) at 550 nm. The Fourier transform infrared (FTIR) spectroscopy was recorded in the range of 400−4000 cm$^{-1}$ at room temperature (Tensor II, Bruker). The stress relaxation tests were conducted by a universal material tester (Zwell/Roell) equipped with a 200 N load cell. The contact angles (CAs) were tested using an optical CA measuring device (OCA20). Thermal stability was analyzed with a TGA thermal analyzer (TA-60WS, Shimadzu), where the specimens (5−8 mg) were filled in an aluminum pan, and then heating from 30 to 800 °C (10 °C/min) under nitrogen atmosphere. The temperature recording was provided by a multichannel temperature tester (T-type thermocouple, JINKO JK804). For differential scanning calorimeter (DSC 8500, Perkin-Elmer), the samples were heated in a ceramic pan at the rate of 20 °C min$^{-1}$ from 25 to 300 °C under nitrogen atmosphere.

### Puncture resistance tests
The puncture resistance tests were conducted on a universal material tester (Zwell/Roell) with the sample-holding apparatus (diameter = 100 mm) and needle (radius = 450 um). The steel needle with the speed of (50 ± 5) mm min$^{-1}$ was used to penetrate the samples for determining the maximum load value.

### Electrochemical and thermosensitive assessment
The change of relative resistance was analyzed by the electrochemical workstation (Autolab). The electronic thermometer and infrared (IR) camera (ST9450) were used to detect the temperature changes. The nonlinear temperature dependent resistance was fitted by Eq. (3) and Eq. (4):

$$R = R_0 + A_0 \exp\left(-\frac{E_a}{2k_B T}\right) \tag{3}$$

$$In(R - R_0) = In(A_0) - \frac{B}{T} \tag{4}$$

where $R_0$ was the resistance of sensor at an infinite temperature, $E_a$ was the thermal activation energy, $k_B$ was the Boltzmann constant, and the term $E_a/2k_B = B$ was the thermal index.

## Data availability

The data generated in this study are provided in the Supplementary Information/Source Data file to ensure access to the minimum dataset. The processed data sets are available as a supplementary dataset excel file to this manuscript. The data generated in this study have been deposited in the public *GitHub* (https://github.com/haosanwei/NC-source-data.git) without any restrictions. Source data are provided with this paper.

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

## Acknowledgements

This work was financially supported by National Natural Science Foundation of China (21978024, 21674013), Beijing Natural Science Foundation (2202034), Fundamental Research Funds for the Central Universities (2021ZY26) and State Key Laboratory of Pulp and Paper Engineering (202212). The authors thank Xi Ying from Shiyanjia Lab (www.shiyanjia.com) for the finite element simulation.

## Author contributions

J.Y. conceived the project. S.H. conducted the majority of experiments and drafted the manuscript. Q.F. and L.M. performed data analysis by the ABSQUS finite element analysis. F.X. and J.Y. participated in experiments and helped revise the manuscript. All authors commented on the manuscript.

## Competing interests

The authors declare no competing interests.
