## [Peer Review File · Nature Communications]

A Biomimetic Laminated Strategy Enabled Strain-interference Free and Durable Flexible Thermistor ElectronicsREVIEWER COMMENTS

Reviewer #1 (Remarks to the Author):

This manuscript describes a nacre-mimetic laminated strategy in combination with topological interlocking and alternating laminated architecture, enabling the in-plane stress dissipation and deformation suppression for achieving high-fidelity temperature discrimination without signal distortion. The details of the investigation are very thorough. However, the use of metal ions for the reinforcement of nacre-mimetic laminated MXene and Cellulose composites is lack of innovation somewhat. Thus, the manuscript has to be carefully revised before being considered for publication.

1. Fig. 4 and Fig. 6 in Supporting Information files are unnecessary because these preparation methods have been widely reported.
2. Figures are blurry; please increase legibility if possible.
3. In Figure 1b (iii), the expression form of the structural formula is better to use the ball and stick model.
4. Page 16, line 377, "(d)" is a clerical error.
5. How is about long-term durability at different temperatures? The authors should supply the stability of $\Delta R/R_0$ under regular changes of several typical temperature for several hours.
6. After the hairdryer treatment with 54.8 °C or place in heating plate at 60 °C, how is about the reusability of TES in practical application.
7. How is about the layer-by-layer assembly and compression assisted technique in improving mechanical property? The stress-strain curves and Young's modulus of TES should be supplied.
8. The strain insensitivity mechanism originated from alternating laminated architecture need to further discussed.

Reviewer #2 (Remarks to the Author):

A high-quality paper should at least reflect two things: novelty and importance, and I am happy that I find both in this work. In this manuscript, the authors depict a universally biomimetic laminated strategy to prepare temperature monitoring sensors that featured fascinating thermosensation ($-1.32\% \text{ } ^\circ\text{C}^{-1}$), superior temperature resolution ($0.3\text{ } ^\circ\text{C}$), and excellent mechanical durability (20000 folding cycles), which can fully activate imaginations and hopes of well-designed flexible thermistor electronics for actual and realistic usages. The key novelty is to achieve the strain interference-free temperature detection and mechanical durability of the flexible thermistor electronics via the nacre-inspired architecture integration, including the compact heterogenous adjacent layers and robust interfacial interlocking. The resultant wearable sensors realize continuous and accurate temperature discrimination without signal distortion under dynamic body motions that is even superior over the bulky commercial thermocouple. More importantly, the experiment and theoretical simulations consistently substantiate that the unique strain deformation interference-free and accurate temperature discrimination originates from the nacre-inspired laminated architecture rather than specific material choices, highlighting the universality of the fabrication method. Overall, the manuscript is well organized, and experimental data are solid with clear explanation, plus pretty nice-looking figure illustration, and the topic would appeal to broad readers. Thus, I enthusiastically recommend its publication in the well-authorized public media of Nature Communications with a few minor suggestions to be implemented by the discretion of the authors.

(1) In Page 6, Line 147-152, "Prior to the following LBL assembly, ...to enhance the PVA crystallinity...". The enhancement of PVA crystallinity is achieved to mimic the robust aragonite platelets in natural nacre. Although the results are discussed in Supplementary Fig. 2 and Fig. 3, a detailed mechanism illustration of this specific process is suggested for the purpose of readily understanding the fabrication of the nacre-like architecture.

(2) It is intuitive that the prepared sensors possess excellent mechanical performances to maintain

structural stability from morphology observation and finite element simulation. However, the benefit of compact nacre-mimetic architecture and interfacial interlocking may have better offer additional clarifications and explanations through mechanical tests, some further discussion can help readers better understand the core conclusions.

(3) The authors emphasize the compact nacre-mimetic architecture of PTF/MXene/Fe composites owing to the abundant interfacial bridging, which is clearly demonstrated as the interlocked adjacent layers in SEM images. Additional experimental evidence such as the comparison of cross-section morphology observation of PVA and PTF/MXene/Fe composites after undergoing large deformation may be a good way to further support the conclusion of efficient interlocking.

(4) In Fig. 5g, the low resistivity pulses of temperature monitoring (40, 50, and 60 °C) upon extreme mechanical impacts (treading by a volunteer and impacting via a hammer) is vividly demonstrated. In order to guarantee the reproducibility, the magnitude of the force is suggested to be considered to define the intensity of external impacts.

(5) The marked peaks in inset of Fig.2c overlay the profiles behind, they should move aside.

(6) Some grammar mistakes should be carefully checked and corrected throughout the manuscript.

(7) The comparison of the prepared thermistor elastomer sensor and commercial thermocouple in Fig.6 reflects the advantages of accurate temperature monitoring without signal distortion. However, the performance and the manufacturer of the commercial thermocouple are not accurately provided, and without this data the competitive advantages of the prepared sensor cannot be rigorously proved.

Reviewer #3 (Remarks to the Author):

Constructing intrinsically stretchable thermistor elastomer sensor, encompassing the excellent thermal resolution and deformability along with obvious resistance signal to accommodate dynamic detection states is of great importance in thermistor electronic applications. Here the authors proposed a nacre-mimetic laminated strategy to realize a versatile MXene-based thermistor elastomer sensor. The authors claim that such sensor delivers competitive advantages of superior thermosensitivity ($-1.32\% \text{ } ^\circ\text{C}^{-1}$), outstanding temperature resolution ($\sim 0.3 \text{ } ^\circ\text{C}$), and unparalleled mechanical durability (20000 folding fatigue cycles), together with considerable improvement in strain-tolerant thermosensation over commercial thermocouple in exercise scenario. Overall the topic presented in this work is interesting, however the fundamental contribution is low since the explanation about the mechanism of the superior properties from structural aspect is not sound. This study in itself did not represent the sufficient novelty of the work, and the comparison with other previous strategies is not sufficient to demonstrate the advantages of the current strategy. In addition, the following comments should be addressed before further considering the paper.

1. The nacre-mimetic laminated strategy is known to be a template to amplify the toughness. The reported thermosensation ($-1.32\% \text{ } ^\circ\text{C}^{-1}$), superior temperature resolution ($0.3 \text{ } ^\circ\text{C}$), and ultrafast responsive time (7 s) seem to mainly depend on the impressive thermosensation mechanism relied on the thermally activated tunnelling current passed through MXene nanosheet junctions as claimed by the authors. The authors need to provide more evidences to clarify the enhanced thermosensation mainly determined by the thermosensitive MXene nanosheets or the nacre-mimetic laminated structure.

2. The authors claim that restraining stress concentration in the proposed biomimetic assemblies can enhance strain insensitivity and overcome the compromising between deformability and thermosensation. Does the nacre-mimetic laminated structure is optimal? Is there any optimal size ratio of the nacreous laminated structure?

3. How do the authors model the mechanical response of TES with alternating laminated architecture with undulating shape under normal stress by using FEM simulation? How do the authors deal with the interface between the laminates? What mechanism can be derived by analyzing the stress field in

the laminated structure?

4. The authors claim that significantly alleviated the strain interference is due to the biomimetic laminated strategy combining with the in-plane stress dissipation and nacre-mimetic hierarchical architecture. However, as is shown in Figure 5 significant relative resistance change due to the applied pressure or strain is still observed. The strain sensitivity did not vanishes. The strain signal decoupling mechanism is not sufficiently clarified in the present manuscript.

5. In addition, how do the authors believe that it is the biomimetic laminated strategy combining with the in-plane stress dissipation and nacre-mimetic hierarchical architecture not other factor that delivers competitive advantages of superior properties?

6. The colorbar shown in Fig.1(d) is unclear.

7. The phrase of “the corresponding in-plane stress diffusion” shown in Fig. 4(d) is unclear.

Point-to-Point Responses to Referees' Comments

We would like to appreciate the three reviewers for their constructive suggestions and insightful comments. Their suggestions help us improve the manuscript's clarity and readability, as well as the rigorous and quantitative aspects of the experiments. Accordingly, we have revised the manuscript throughout by including additional experimental results and discussion. Overall, the revised manuscript contains 31 new figures that address the queries of the three reviewers. We feel that the revised work is significantly improved as a result of these constructive insights. The point-to-point responses to the reviewers' comments are detailed below (black color for responded texts to the reviewers, and red color for the revised texts).

Reviewer #1 (Remarks to the Author):

This manuscript describes a nacre-mimetic laminated strategy in combination with topological interlocking and alternating laminated architecture, enabling the in-plane stress dissipation and deformation suppression for achieving high-fidelity temperature discrimination without signal distortion. The details of the investigation are very thorough. However, the use of metal ions for the reinforcement of nacre-mimetic laminated MXene and Cellulose composites is lack of innovation somewhat. Thus, the manuscript has to be carefully revised before being considered for publication.

Response: We sincerely appreciate the careful reading of the manuscript. The main criticism of the reviewer is that the novelty of interlocking the alternating laminated architecture via synergistic effects among MXene nanosheets, metal ions, and cellulose composites has been well-defined. Here we would like to highlight the novelty of this work from the following three aspects.

Response to the novelty of this study:

(1) The demonstration of a universal and customizable interlocking laminated strategy for strain interference-free temperature monitoring electronics.

The most prominent advantage of our work is to successfully solve the sensing cross-talk conundrum when the thermistor elastomer sensor (TES) subjected to the temperature-strain dual stimuli (**Supplementary Movie 4**), and this ability to differentiate multiple stimuli combined with excellent mechanical durability is of profound importance for satisfying the accurate temperature indication requirements of advanced e-skin systems towards the human-machine interaction, healthcare monitoring, and soft robotics.

It is undeniable that the numerous nacre-like structural designs and nacre-inspired layered nanocomposites have been well developed for excellent mechanical performances or other properties integration (such as electromagnetic interference shielding and thermal management) because of the heterostructure and multilayer characteristic. However, natural wisdoms of the "brick-and-mortar" structural features of natural nacre (**Fig. R1**) are still underused.

Fig. R1. Natural nacre from abalone shell (inset), showing the lamellar microstructure (Guo et al., Nat. Mater. 2022. <https://doi.org/10.1038/s41563-022-01292-4>).

Most of the nacre-like materials mainly focus on mimicking the laminated architecture, but neglect the efficient interfacial bridging for preventing the adjacent lamellar dislocation motion and crack propagation that shows great potential for maintaining the stable conductive path between the adjacent layer barriers and facilitates the in-plane stress dispersion, enabling the synergistic toughening and strain insensitivity. Even though some of reports achieved the interfacial cross-linking via the relatively weak dynamic interactions (e.g. hydrogen bonds) with the assistance of vacuum filtration or hot pressing technique, it is still hard to realize the invariable mechanical functions through remarkable interlayer coupling, effective stress transfer, and twisted crack propagation. We compare the preparation routes with previous reports that possessed similar nacre-mimetic structure without efficient interlocking, aiming at demonstrating the simplicity and universal interlocking of proposed nacre-mimetic laminated architecture (**Fig. R2**).

Our work:

Adv. Funct. Mater. 2021, 32, 2110782

Fig. R2. The comparison of our strategy with the reported work for the synthesis route of nacre-mimetic architecture in literature.

In our work, an effective and universal coordination bond between carboxyl groups and metal ions is deliberately sought to act as robust interfacial bridging for coupling adjacent layers that has never been reported in nacre-mimetic architecture. Additionally, the achieved strain insensitivity originates from interlocking laminated architecture rather than the deliberate material compositions consideration, which can be facily extended to various nanofillers including carbon black, graphene, carbon nanotube, and polyaniline (**Supplementary Fig.32**), inferring the extraordinary universality and customizability of the proposed strategy. Indeed, the simplicity and efficient interfacial crosslinking via interlayer spraying associated with the layer-by-layer assembly technique also promotes the scalable fabrication ($25 \times 45 \text{ cm}^2$) as demonstrated in **Supplementary Fig. 10a**, showing the scalable prospect for practical

application. We foresee that the universal nacre-mimetic architecture together with simplicity interlocking approach offers a myriad of opportunities to facily design polymer-based thermistor sensor through numerous combinations of various thermosensation composites, thus will drive the innovation of flexible electronics and promote the creation of advanced nacre-mimetic materials.

Supplementary Fig. 10a. Optical photograph of the resultant large-area ($25 \times 45 \text{ cm}^2$) PTF/MXene/Fe composites by LBL assembly, showing favorable scalability.

(2) An outstanding combination of mechanical and thermosensation properties.

In this work, we address the long-standing issue of strain interference in thermistor electronic applications combined with excellent mechanical and thermosensation properties in terms of superior thermosensitivity ($1.32 \% \text{ } ^\circ\text{C}^{-1}$), extraordinary resolution ratio ($0.3 \text{ } ^\circ\text{C}$), together with unprecedented sensing durability that capable of temperature acquisition after 20000 folding fatigue cycles and even subjected to harsh mechanical shock. The substantial progress of unparalleled temperature discrimination without signal distortion is achieved that even competitive to the commercial thermocouple (T-type, JINKO JK804) in exercise scenario (**Fig. 6e-c**), thus expected to lead the development of advanced biomimetic materials for flexible and wearable applications. We provide a summary picture to fully reflect the novelty in the both aspect of architecture and performances (**Supplementary Table3**), and the comparison with previous sensors based on various structural strategies is also presented in **Supplementary Fig. 37**. We firmly believe that this is the first demonstration of the thermistor elastomer sensor that simultaneously achieves a wide set of demanding properties with a high level of success for dynamic scenario.

Supplementary Fig. 37. The summary of structural characteristics and beneficial features of the proposed biomimetic laminated strategy.

(3) Comparison with previous reports in literature and the contribution of our work.

Another major contribution in this study is the mechanism demystification by theoretical model simulation, microstructure observation, and superposed signal detection. Compared to previous approaches to produce nacre-mimetic materials with “planar lamellar” model, we verify that the synergistic effects between the interlocking enhanced in-plane stress dispersion and the layer barrier weakened structural integrity, simultaneously result in the optimization of mechanical resilience and structural stability. Therefore, our work can provide new insights into the key role of the interfacial bridging rather than the “planar lamellar” model on the property optimization of nacre-like materials. We further supplement a comparison of our work with previously reported nacre-mimetic strategy in literature to clarify the unique contribution especially in terms of putting forward the interfacial bridging and revealing the in-plane stress dispersion mechanism (**Table R1**).

Table R1. Comparison of the novelty of the interlocking laminated strategy in this work with counterparts based on similar nacre-mimetic architecture.

Refs.	Components	Method	Structure	Microscopic observation	Interfacial bridging	Mechanism explanation	Application
1	Nano-clay/BC	Hot pressing	Nacre-mimetic		Hydrogen bonding	N/A	Plastic substitute
2	MXene/sodium carboxymethyl cellulose	Vacuum-assisted filtration	Nacre-mimetic		Hydrogen/ covalent bonding	N/A	Electromagnetic interference shielding
3	Polyacrylonitrile/ rGO	Electrospinning carbonization techniques	Nacre-mimetic		N/A	Finite element simulation	Flexible electronics
4	MXene	Vacuum-assisted filtration	Nacre-mimetic	(d) 	Hydrogen bonding/ van der Waals forces	N/A	Electromagnetic interference shielding
5	MXene/PVA	Vacuum-assisted filtration	Nacre-mimetic		Hydrogen bonding	Finite element simulation	Flexible electronics
6	Phenylphosphonic acid/graphene nanoplatelets/poly(vinyl alcohol)	Layer-by-layer assembly/ hot pressing	Nacre-mimetic		Hydrogen-bonding/ π-π stacking	N/A	Electromagnetic interference shielding
7	GO / leaf-like MnO ₂ hexagon nanosheets	Layer-by-layer assembly/ hot pressing	Nacre-mimetic		Non-covalent/ covalent bonding	Finite element simulation/ Molecular Simulation	Aerospace, biomedicine and electronics
Our work	PVA/MXene/ TOCNF/Fe	Layer-by-layer assembly/ hot pressing/ Interlocking	Nacre-mimetic		Hydrogen/ coordination bonding	Finite element simulation	Flexible thermoistor electronics

- [1] Guan, Q.-F., Ling, Z.-C., Han, Z.-M., Yang, H.-B., Yu, S.-H., Ultra-strong, ultra-tough, transparent, and sustainable nanocomposite films for plastic substitute. *Matter* **3**, 1308-1317 (2020).
- [2] Wan, S. J., et al. High-strength scalable MXene films through bridging-induced densification. *Science* **374**, 96-99 (2021).
- [3] Zan, G. T., et al. A biomimetic conductive super-foldable material. *Matter* **4**, 3232-3247 (2021).
- [4] Liu, J., et al. Hydrophobic, flexible, and lightweight MXene foams for high-performance electromagnetic-interference shielding. *Adv. Mater.* **29**, 1702367 (2017).
- [5] Zhao, L. J., et al. Highly-stable polymer-crosslinked 2D MXene-based flexible biocompatible electronic skins for in vivo biomonitoring. *Nano Energy* **84**, 105921 (2021).
- [6] Chen, Q., et al. Scalable, Robust, Low-Cost, and Highly Thermally Conductive Anisotropic Nanocomposite Films for Safe and Efficient Thermal Management. *Adv. Funct. Mater.* **32**, 2110782 (2022).
- [7] Chen, K., et al. Graphene oxide bulk material reinforced by heterophase platelets with multiscale interface crosslinking. *Nat Mater* (2022): 1-9 <https://doi.org/10.1038/s41563-022-01292-4>.

1. Fig.4 and Fig.6 in Supporting Information files are unnecessary because these preparation methods have been widely reported.

Response: Thanks for suggestion, and the **Supplementary Fig.4** and **Fig.6** have been removed from Supplementary Information.

2. Figures are blurry; please increase legibility if possible.

Response: Thanks for suggestion and we improve the quality of the figures in the manuscript to improve the clarity and readability.

3. In Figure 1b (iii), the expression form of the structural formula is better to use the ball and stick model.

Response: We appreciate the reviewer's comments. As illustration in **Fig.R3**, we find that the ball-and-stick model of functional groups (-OH, -F, and -COOH) at molecular scale cannot present the TOCNF and MXene nanosheets model in a vivid manner, and it is difficult to give a reasonable illustration of the arrangement without additional supplementary texts in **Fig. 1b (iii)**, we thereby prefer to use the original presentation structural formula. By your suggestion, the more detailed interfacial bridging demonstration by total ball and stick model is given in **Fig.2j**.

Fig. R3

Fig. 2j

Fig.R3. The proposed interfacial bridging between Fe(II) and functional groups (-OH, -COOH, and -F) by the ball-and-stick model and physical model in both molecular and microscale. **Fig.2j.** Illustration of interfacial bridging (coordination bonds, hydrogen bonds, and ionic interactions) between adjacent layers for topological interlocking.

4. Page 16, line 377, “(d)” is a clerical error.

Response: Sorry for the careless and we correct the “(d)” for “(e)” in **Fig.4**.

5. How is about long-term durability at different temperatures? The authors should supply the stability of $\Delta R/R_0$ under regular changes of several typical temperature for several hours.

Response: Thanks for this valuable question to verify the long-term lifespan of the thermistor elastomer sensor (TES). We have investigated the sensing durability (**Supplementary Fig.17**) of prepared TES at different temperatures (30, 40, and 45 °C) that find the negligible deviation under each temperature, implying excellent durability and stability of the resultant TES. Meanwhile, consecutive temperature identification up to 100 cycles from 36-38 °C to mimic the continuous “fever indication” is presented in **Fig.6d**, and it not only achieves the comparable durability with the commercial thermocouple (T-type, JINKO JK804), but also exceeds the most of the reports in literature (e.g. Adv. Funct. Mater. 2021, 2107570; ACS Nano 2020, 14, 218–228; Adv. Sci. 2021, 8, 2004377; Adv. Funct. Mater. 2021, 31, 2007661; Nano Lett. 2020, 20, 6176–6184).

Notably, water as the heat transfer medium is used in the temperature measurement to allow the rapid conversion of the two temperature environments (**Supplementary Fig. 11a**). However, it is difficult for us to provide two constant temperature conditions in the laboratory up to several hours without generating environmental heat exchange because of the thermal diffusion effect (**Supplementary Fig. 11b**), and it is hard to monitor a further long-term temperature test via consistent monitoring frequency due to lack of corresponding test instruments. Therefore, we hope the reviewer can agree with us to extend the durability test up to 100 cycles.

Updated Supplementary Fig. 11 in Supplementary Information

Supplementary Fig. 11. a Schematic diagram of thermosensation measurement using water as the heat transfer medium. b Scheme of thermal diffusion effect in continuous temperature measurement.

As for the stability concern, we investigate the time gradient influence on the reliability of TES, and the stable signal output still could be recorded and possesses 90.9% TCR value retention even over a long period of 45 days that sufficient for the most of possible application duration, corroborating the extraordinary long-term reliability of the TES (**Supplementary Fig. 18**).

Updated Supplementary Fig. 18 in Supplementary Information

Supplementary Fig. 18. a Long-term operation of the TES over 45 days in the body temperature range of 30-40 °C. **b** The subtle TCR variation of TES with different contents of MXene nanosheets over 45 days.

6. After the hairdryer treatment with 54.8 °C or place in heating plate at 60 °C, how is about the reusability of TES in practical application.

Response: The TES shows a monotonic decrease in electrical resistance with increasing temperature from 20-80 °C (**Fig. 3g**), demonstrating its potential for broad application beyond human body temperature monitoring. To prove the reliability thermosensation for high temperature application, the durability of TES is evaluated by cyclic heating and cooling under hairdryer (**Supplementary Fig. 31**). The TES maintains the almost identical $\Delta R/R_0$ curves without marginal deviation from linearity over 70 cycles under hairdryer heating flow (54.8 °C). The results further confirm the stability of the TES in the wide temperature range, exhibiting promising application prospect.

Updated Supplementary Fig. 31 in Supplementary Information

Supplementary Fig. 31. $\Delta R/R_0$ curves of the TES under hairdryer heating flow (54.8 °C) for 70 cycles, showing excellent durability for high temperature monitoring.

7. How is about the layer-by-layer assembly and compression assisted technique in improving mechanical property? The stress-strain curves and Young's modulus of TES should be supplied.

Response: The reviewer's suggestion on investigating the mechanical performances of TES could help better elucidate the interlocking laminated designs and inspires us to verify that the nacre-mimetic heterostructure and interfacial bridging enables excellent mechanical durability (20000 cycles, **Fig. 5h**) towards the repeated folding in daily wearable application. In addition to answering the reviewer's concerns about the stress-strain curves and Young's modulus of TES, we offer a comprehensive mechanical testing section for clarifying the structural optimization, nanofiller's reinforcement, and tough interfacial interlocking. The main supplementary data and discussed mechanical performances as follows:

Updated Supplementary Note 2 in Supplementary Information:

Although the large-size PTF/MXene/Fe composites (sensing layer, 25 × 45 cm²) can be fabricated via the straightforward interfacial bridging and facile LBL assembly procedure towards the commercial application (**Supplementary Fig. 10a**), the basic mechanical requirements of FTEE withstanding arbitrary deformations requires further structural optimization for mechanically resilience and robustness, which weighs equal significance compared to the thermosensitivity. For this concern, the influence of the stacked layers number (1, 4, 8, and 16) on mechanical performance is initially discussed owing to its preliminary role in balancing the stiffness and elasticity (**Supplementary Fig. 10b**), and the sensing layer demonstrated the dramatic improvement in robustness by readily altering the layer number (the corresponding thickness from 0.12 to 0.99 mm) that attained the maximum tensile strain and stress of 1694.8 % and 5.76 MPa, respectively. Considering the excessive layers (thickness increasing) inevitably compromise the mechanical harmony with skin because of the excessive rigidity, the

eight-layer of alternating laminated architecture is chosen as the optimized configuration.

Additionally, the uniaxial tensile tests corroborate that the proportion of TOCNF (from 0.25 to 1.25 wt%) is crucial to further optimize the mechanical properties of PTF/MXene composites across a wide range including tensile stress from 3.38 to 6.68 MPa and Young's modulus from 0.11 to 0.39 MPa, respectively (**Supplementary Fig. 10c**). Simultaneously, the pronounced increasing of elasticity with the addition of Fe(II) is demonstrated for all counterpart samples, convincing the desirable reinforcement effect of the coordination bonds (**Supplementary Fig. 10d**). In view of the skin-like Young's modulus (0.35 ± 0.032 MPa) and a relatively high toughness (50.62 ± 1.53 MJ m⁻³), the TOCNF content of 1 wt% is optimal for the subsequent context.

Moreover, the PTF/MXene/Fe composites can be easily elongated to more than 10 times its original length without fracture by uniaxial test, revealing excellent elasticity and stretchability (**Supplementary Fig. 10e and 10f**). Compared with the PVA composites, the mechanical performances of PTF/MXene/Fe composites were significantly improved, including tensile stress (from 3.38 to 6.68 MPa), elastic modulus (from 0.11 to 0.39 MPa), and toughness (from 28.58 to 56.98 MJ m⁻³). This phenomenon could be mainly attributed to the stiffness of individual TOCNF and dynamic interactions (hydrogen bonds) among TOCNF, MXene nanosheets, and polymer matrix, leading to synergistic effects on the mechanical resilience. Besides, the unique interfacial bridging of coordination bonds between -COOH groups and Fe(II) contribute to the increasement in both stiffness and toughness without sacrificing extensibility.

Updated Section S3 Mechanical performances in Supplementary Information:

Supplementary Fig.10 Mechanical optimization. **a** Optical photograph of the resultant large-area (25 × 45 cm²) PTF/MXene/Fe composites by LBL assembly, showing favorable scalability. **b** The laminated layer number (1, 4, 8, and 16 layers, respectively) dependent tensile stress–strain curves of pristine PVA with alternating laminated architecture. **c** The typical tensile stress–strain curves of PTF/MXene and PTF/MXene/Fe composites with different TOCNF contents (0.25, 0.5, 0.75, 1, and 1.25 wt%, respectively), and **d** the corresponding TOCNF content dependent Young's modulus and tensile stress, error bars were defined as S.D. (n = 3 independent samples). **e** Uniaxial tensile test of PVA, PTF, PTF/MXene, and PTF/MXene/Fe composites, respectively, and **f** the corresponding mechanical parameters including Young's modulus, toughness, tensile strain, and tensile stress.

8.The strain insensitivity mechanism originated from alternating laminated architecture need to further discussed.

Response: Thanks for your constructive comment to help us improving the manuscript. The strain insensitivity that correlated with structural changes is mainly ascribed to the compact and interlocked heterostructure. This feature is clearly reflected in mechanical experiments (e.g. puncture resistance tests, tear tests, and cyclic tensile tests) and microstructure observation (SEM images and EDS mapping) as mentioned by reviewers (all supplemented in the revised manuscript), which could be used as direct evidence to further prove the advantage of interfacial bridging between heterogenous adjacent layer and clarify the strain-unperturbed mechanism. Moreover, we supplement some new results to demonstrate the benefits of interlocking laminated architecture and further discuss the strain-unperturbed mechanism as follows:

(1) The outstanding puncture resistance showing efficient interfacial bridging.

As illustrates in **Fig. 5** and **Supplementary Fig. 27**, the PTF/MXene/Fe composites with thickness of only 0.60 mm could withstand a puncture force as high as 13.42 N. Comparatively, the PVA, PTF, and PTF/MXene composites are pierced easily with a much smaller displacement where the force is only 3.9, 6.7, and 7.6 N, respectively. This striking out-plane puncture tolerance reflects the dominant role of coordination bonds in topological interlocking where the efficient stress dissipation is achieved through the interfacial bridging so as to enable excellent impact resistance. In contrast, the poor interlayer bonding (e.g. the bonding in PVA, PTF, and PTF/MXene) without efficient stress transfer upon engaged with mechanical shock faces with stress concentration, resulting in the separation and structural destruction. Indeed, the energy dispersive spectrometer (EDS) mapping in **Supplementary Fig. 28** provides more clues that the Fe element as the interfacial bridging agent predominantly exists in the adjacent layers, confirming our assumption that coordination bonds playing an indispensable role in facilitating the stress dispersion.

Updated Fig. 5 and Supplementary Fig. 28 in Manuscript and Supplementary Information:

Fig.5 Puncture resistance test (needle diameter = 900 μm , speed = 50 mm/min) of PVA, PTF, PTF/MXene, and PTF/MXene/Fe, respectively, and schematic diagram of the topological interlocking for eliminating the interface failure and dissipating energy upon the loading.

Supplementary Fig.28. The EDS mapping images for uniform distribution elements of C, O, and Ti, and the detailed spatial distribution of Fe as interfacial bridging agent.

(2) The comparison of strain insensitivity for TES with/without interlocking laminated architecture.

To further validate the roles of the key design features in strain insensitivity, i.e., the interlocking and heterogeneous adjacent layer barrier, we fabricate and examine a type of reference sensor (defined as TES with homogeneous structure) without one of these features through the same preparation steps (keeping the TOCNF, MXene nanosheets, and Fe contents), only eliminating layer-by-layer assembly and interlayer spraying processes.

As shown in **Fig. 5a**, the pressure sensing behaviors of integrated TES without interlocking laminated architecture show good linearity over two continuous pressure ranges with sensitivity of 1.53 kPa^{-1} from 0 to 12 kPa and 0.088 kPa^{-1} from 12 to 100

kPa, respectively. In contrast, the TES with interlocking laminated architecture presents a much weakened piezoresistive effect (0.66 and 0.075 kPa^{-1} in both 0 - 20 kPa of low-pressure and 20 - 100 kPa of high-pressure ranges). In particular, the pressure sensitivity is reduced about 2.3 times and the sensing range is extended from 0 - 12 to 0 - 20 kPa in the first piezoresistive response interval.

This comparative result further clarifies the mechanism as we mentioned in the manuscript that the heterogeneous adjacent layers are coupled by effective interfacial bridging, either already under physical contact or only marginally separated to hinder the creation of new conduction paths. Upon the external loading, the strong interfacial bridging alleviates the stress concentration for in-plane stress dissipation, giving rise to the deformation suppression and reveals as a very limited $\Delta R/R_0$ variation for preventing the strain induced signal distortion.

Besides, the TES also presents a strain-unperturbed sensing performance in uniaxial stretching (**Fig. 5d**). Compared to the TES with homogeneous structure, the linear resistance variation of TES with interlocking laminated architecture is decreased by 17.92% over the entire skin stretching range (0 - 100%), together with a limited detection sensitivity ($GF \sim 0.467$), owing to the conductive path barrier and the restrained dislocation motion and crack propagation via interlocking laminated architecture. As a result, both the strain and pressure sensitivity are far below the level of most previous flexible electronics in literature as we listed (**Table R2**).

Updated Fig. 5a and d in Manuscript:

Fig. 5 a The weak piezoresistive effect in the applied pressure region (0 - 100 kPa), and **d** unobvious gauge factor (GF) in 100% strain range for TES with interlocking laminated architecture in contrast to the TES without structural designs.

Table R2. Comparison of the TES in this work with previous flexible electronics in literature, showing obvious strain and pressure insensitivity.

Refs.	Materials	Strain sensitivity (GF)	Working range	Interference-free monitoring
Small 2021 , 2100542	Silver nanowires/ reduced graphene oxide/ thermoplastic polyurethane	57.5 for 0-111% 472.9 for 111-167% 1902.5 for 167-200%	0-200 %	No
Adv. Funct. Mater. 2021 , 31, 2007661	Black phosphorus/ laser-engraved graphene	81 for 0-8% 303 for 8-20% 2765 for 20-25%	0-20%	Yes
npj Flexible Electronics. 2022 , 6,11	MXene/ 3, 4-ethylene dioxithiophene	87.15 for 0-13% 413.36 for 13-22.5% 2074.95 for 22.5-25%	0-25%	Yes
ACS Nano 2020 , 14, 218-228	PAA/PANI	18.28 for 0-268.9 %	0-268.9 %	No
Adv. Funct. Mater. 2021 , 31, 2010465	Polydopamine/ graphene oxide/ cellulose	0.73 for 0-10% 1.18 for 10-15% 1.76 for 15-20%	0-20%	No
Sci. Adv. 2020, 6, eabb5367	MXene-PpyNWVSNP-PAM	16.9 for 0-800% 11.2 for 800-2800%	0-2800%	No
Adv. Funct. Mater. 2020 , 2005135	PAM/CNF/MXene	4.15 for 0-250% 8.21 for 250-500%	0-500%	No
Our work	PVA/TOCNF/MXene/Fe	0.467 for 0-100 %	0-100 %	Yes

Refs.	Materials	Pressure sensitivity	Working range	Interference-free monitoring
ACS Nano 2021 , 15, 9746- 9758	MXene/silk fibroin	298.4 kPa ⁻¹ for 1.4-15.7 kPa 171.9 kPa ⁻¹ for 15.7-39.3 kPa	0-39.3 kPa	No
Nat Commun 2022 , 13,1119	MXene	1928.8 kPa ⁻¹ for 0-0.02 Pa	0-1 Pa	No
Nano Energy 2022 , 95, 106986	MXene/GO/PS	115 kPa ⁻¹ for 0-7.58 kPa 224 kPa ⁻¹ for 7.58-20.65 kPa	0-20 kPa	No
Adv. Sci. 2022 , 2200507	MXene/PU	281.54 kPa ⁻¹ for 0.2-1.7 kPa 509.78 kPa ⁻¹ for 1.7-5.7 kPa 66.68 kPa ⁻¹ for 5.7-20.3 kPa	0-20 kPa	No
ACS Nano 2020 , 14, 2145-2155	MXene/PDMS	151.4 kPa ⁻¹ for 0-4.7kPa 33.8 kPa ⁻¹ for 4.7-15 kPa	0-15 kPa	No
Nano Lett. 2022 , 22, 4459-4467	MXene/PEO	777 kPa ⁻¹ for 0-20 Pa	0-20 Pa	No
Matter 2022 , 5, 1-21	CNT/PEDOT:PSS	126135.9 kPa ⁻¹ for 0-1.2 kPa 291699.6 kPa ⁻¹ for 1.2-12 kPa 43842.1 kPa ⁻¹ for 12-20 kPa	0-20 kPa	No
Our work	PVA/TOCNF/MXene/Fe	0.66 kPa⁻¹ for 0-17.76kPa 0.075 kPa⁻¹ for 17.76-100 kPa	0-100 kPa	Yes

(3) Validation of the strain-unperturbed mechanism from the designed structure and finite element simulation.

From a more quantitative analysis, we also confirm that the strain-unperturbed performances of TES can be optimized by tailoring the parameters like the layer number of nacre-mimetic architecture (**Fig. 5e**). Typically, the TES shows diminishing strain sensitivity from 0.89 to 0.36 with the increasing of the layer number (1-16) in the deformation rang of 1-100 %, meaning that the multiple adjacent layer barrier and interface crosslinking synergistically contribute to the toughening mechanism (demonstrated as the increased stress at 100 %) and suppress the crack propagation along the deformation direction, which act as the structural variables for regulating the strain insensitivity.

In order to further elucidate the mechanism, the FEM analysis is conducted to show the indispensable role of our strategy by exploring the stress distributions in the architecture upon external loading. When the architecture is stressed, the dramatically contact area increasing of adjacent layer interface would significantly lower the resistance variation. Since the topological interlocking as the “interface bridging” further promotes the stress transfer that demonstration as the color diffusion from in-plane view along with the increasingly stepwise-pressure-applied stress distribution, inferring the structural variation mediated electronic output has been suppressed (**Supplementary Fig. 24c**).

Overall, we report the design of a flexible thermistor elastomer sensor that achieves intrinsically strain-unperturbed temperature monitoring in conjunction with the impressive mechanical performances including stretchability, bending durability, and puncture resistance. This strain-unperturbed mechanism is comprehensively revealed by mechanical tests, microstructure observations, quantitative structural optimization, and finite element simulation in our revised manuscript. Specifically, the use of interfacial bridging combined with hot pressing technique formed the compact interface for the sensor’s overall nacre-mimetic architecture that dominated by the topological interlocking, which promotes the in-plane stress dissipation rather than the total structural variations under the external loading. By virtue of the adjacent layer barrier of heterostructures suppress the formation of conductive paths, together with the assistance from interface crosslinking, our strategy further affects dislocation motion and crack propagation, and achieves the weak piezoresistive effect and strain insensitivity.

Updated Fig. 5e in Manuscript:

Fig. 5. e The correlation of sensing layers numbers of TES with strain sensitivity and stress at 100% deformation, along with the thickness increasing.

Supplementary Fig. 24. c Finite element simulation results of TES form in-plane view for stress diffusion at different deformation ratios.

Reviewer #2 (Remarks to the Author):

A high-quality paper should at least reflect two things: novelty and importance, and I am happy that I find both in this work. In this manuscript, the authors depict a universally biomimetic laminated strategy to prepare temperature monitoring sensors that featured fascinating thermosensation ($-1.32\% \text{ } ^\circ\text{C}^{-1}$), superior temperature resolution ($0.3\text{ } ^\circ\text{C}$), and excellent mechanical durability (20000 folding cycles), which can fully activate imaginations and hopes of well-designed flexible thermistor electronics for actual and realistic usages. The key novelty is to achieve the strain interference-free temperature detection and mechanical durability of the flexible thermistor electronics via the nacre-inspired architecture integration, including the compact heterogenous adjacent layers and robust interfacial interlocking. The resultant wearable sensors realize continuous and accurate temperature discrimination without signal distortion under dynamic body motions that is even superior over the bulky commercial thermocouple. More importantly, the experiment and theoretical simulations consistently substantiate that the unique strain deformation interference-free and accurate temperature discrimination originates from the nacre-inspired laminated architecture rather than specific material choices, highlighting the universality of the fabrication method. Overall, the manuscript is well organized, and experimental data are solid with clear explanation, plus pretty nice-looking figure illustration, and the topic would appeal to broad readers. Thus, I enthusiastically recommend its publication in the well-authorized public media of Nature Communications with a few minor suggestions to be implemented by the discretion of the authors.

Response: We thank the positive and encouraging comments by the reviewer, and we are equally inspired by the kind suggestions from the reviewer. The suggestions have improved the quality of the manuscript with respects to the morphological characterization and mechanical performances of the thermistor elastomer sensor.

1. In Page 6, Line 147-152, "Prior to the following LBL assembly, ...to enhance the PVA crystallinity...". The enhancement of PVA crystallinity is achieved to mimic the robust aragonite platelets in natural nacre. Although the results are discussed in Supplementary Fig. 2 and Fig. 3, a detailed mechanism illustration of this specific process is suggested for the purpose of readily understanding the fabrication of the nacre-like architecture.

Response: We appreciate the reviewer for suggesting the addition of further mechanism exploration of improving PVA polymer crystallinity, and this could indeed facilitate the explanation of the construction of robust nacre-mimetic architecture. As detailed of interpretation in **Supplementary Note 1**, applying a strong alkaline hydroxide treatment of dried PVA film results in the following two aspects. First, OH^- of the alkaline hydroxide attacks the hydroxyl groups of PVA, resulting in the disrupted hydrogen bonds and deprotonation of the hydroxyl groups of the PVA chain. Afterward,

the newly formed O- groups in PVA interact with free Na⁺ ions to form complexation. These two sequential processes facilitate the PVA crystallization and they are depicted in revised **Supplementary Fig. 2**.

Updated Supplementary Fig. 2 in the Supplementary Information:

Supplementary Fig. 2. Schematic illustration of preparation and proposed mechanism of high crystallinity PVA polymer networks.

2. *It is intuitive that the prepared sensors possess excellent mechanical performances to maintain structural stability from morphology observation and finite element simulation. However, the benefit of compact nacre-mimetic architecture and interfacial interlocking may have better offer additional clarifications and explanations through mechanical tests, some further discussion can help readers better understand the core conclusions.*

Response: We are glad that the reviewer shares the idea that the additional mechanical tests will be a good manner to further improve the clarity and reliability of the interfacial bridging for alternating laminated architecture that enables the mechanical resilience and strain insensitivity under deformation conditions.

To further justify this point, we conduct some additional discussion on mechanical performance via puncture resistance and tearing tests (see **Experimental Methods** for more details). Typically, the PTF/MXene/Fe composites of sensing layer (0.60 mm) can withstand a puncture force (needle diameter = 900 μ m, loading speed = 50 mm/min) as high as 13.42 N, which is superior to PVA, PTF, and PTF/MXene composites with the force only 3.9, 6.7, and 7.6 N, respectively (**Fig. 5c** and **Supplementary Fig. 27**). This striking out-plane puncture tolerance further reflects the dominant role of coordination bonds in topological interlocking where the efficient stress dissipation is achieved through the interfacial bridging so as to enable excellent impact resistance. Likewise, with the elongation of notched rectangular PTF/MXene/Fe composites specimen (size = 10 \times 5 mm, stretching speed = 3 mm/min), the crack gradually widens along the longitudinal direction until the sample becomes damaged at a strain of 1398%

(Supplementary Fig. 28a and 28b). The fracture energy of the PTF/MXene/Fe composites is calculated to be as large as $591 \pm 35 \text{ kJ m}^{-2}$, which is over 1.3 times higher than that of PVA ($454 \pm 24 \text{ kJ m}^{-2}$), because of the presence of interfacial interlocking and dynamic cross-linking in the matrix (Supplementary Fig. 28c).

From the above analyses, the integrated TOCNF units combined with polymer-based crosslinkers in nacre-mimetic architecture provide an intrinsic strengthening and toughening mechanism, enhancing the mechanical properties of the PTF/MXene composites. Moreover, the efficient interlocking of coordination bonds between mechanically optimized lamellae is beneficial for effective loading transfer within the whole system, improving the holistic mechanical performance, especially demonstrated as the outstanding puncture resistance that is consistent with the finite element simulation.

Updated Fig.5c in Manuscript and Supplementary Fig.27 in Supplementary Information:

Fig. 5c

Fig. 27

Fig. 5c The puncture resistance curves for PVA, PTF, PTF/MXene, and PTF/MXene/Fe composites, respectively. Supplementary Fig. 27 Photographs of puncture resistance test and the corresponding needle size parameters (needle diameter = 900 μm, loading speed = 50 mm/min).

Updated Supplementary Fig.28 in Supplementary Information:

Supplementary Fig. 28. Typical stress-strain curves of the unnotched and notched a PVA and b PTF/MXene/Fe composites, the inset photograph of the elongation for notched composites and the formula of tearing energy calculation. c Tearing energy of PVA and PTF/MXene/Fe composites.

3. The authors emphasize the compact nacre-mimetic architecture of PTF/MXene/Fe composites owing to the abundant interfacial bridging, which is clearly demonstrated as the interlocked adjacent layers in SEM images. Additional experimental evidence such as the comparison of cross-section morphology observation of PVA and PTF/MXene/Fe composites after undergoing large deformation may be a good way to further support the conclusion of efficient interlocking.

Response: Thank you for evaluating our study in-depth and points out such meticulous advice. We have added the cyclic loading–unloading tensile curves (100 cycles, 500 % strain) for PVA and PTF/MXene/Fe composites, and the effective interlayer bridging is proved by comparing the interface variations in microscopic structure observation after cyclic fatigue test.

Typically, a pronounced hysteresis loop with the larger residual strain is noted from the successive loading–unloading curves in comparison with that of PVA, resulting in the exceptional recovery efficiency ($\sim 73.57\%$) even after 100 fatigue cycles at strain of 500 % (Fig. 4d). The inset cross-section SEM images further demonstrate the dense interface of PTF/MXene/Fe composites in contrast to the delamination and slippage between adjacent layers of the PVA that are negatively characterized by the appearance of extra interlayer voids, inferring the interface failure and weak fatigue resistance. These results verify the assumption that the dynamic interactions and coordination bonds induce effective interlayer coupling and facilitate the stress dissipation to the in-plane dynamic crosslinking networks for alleviating the high-stress region concentration, thus agrees well with the simulated data.

Updated Fig 4.d in the Manuscript:

Fig. 4d Cyclic loading–unloading tensile curves (100 cycles, 500 % strain) for PTF/MXene/Fe and PVA composites, the inset showing the cross-section SEM images after 100 stretching cycles.

4. In Fig. 5g, the low resistivity pulses of temperature monitoring (40, 50, and 60 °C) upon extreme mechanical impacts (treading by a volunteer and impacting via a hammer) is vividly demonstrated. In order to guarantee the reproducibility, the magnitude of the force is suggested to be considered to define the intensity of external impacts.

Response: We agree with this comment. We have revised the corresponding texts in the manuscript for reproducibility and clarity.

5. The marked peaks in inset of Fig.2c overlay the profiles behind, they should move aside.

Response: Sorry for this carelessness.

Updated Fig. 2c in the Manuscript:

6. Some grammar mistakes should be carefully checked and corrected throughout the manuscript.

Response: Thanks for your patience to read throughout our manuscript, and we have carefully read throughout the manuscript and try our best to improve the language.

7. The comparison of the prepared thermistor elastomer sensor and commercial thermocouple in Fig.6 reflects the advantages of accurate temperature monitoring without signal distortion. However, the performance and the manufacturer of the commercial thermocouple are not accurately provided, and without this data the competitive advantages of the prepared sensor cannot be rigorously proved.

Response: We thank the reviewer for the insightful comments. To exclude such vagueness, the parameters of commercial thermocouple are marked in detail in revised manuscript.

Updated texts in the Manuscript:

- (1) With regard to further confirming the application potential, several competitive advantages including impressive temperature resolution, long-term lifespan, fast response time, and precise temperature detection are given priority of consideration with the comparison of commercial T-type thermocouples (diameter = 0.3 mm, resolution =0.2% +1°C).
- (2) The temperature recording was provided by a multichannel temperature tester (T-type thermocouple, JINKO JK804).

Reviewer #3 (Remarks to the Author):

Constructing intrinsically stretchable thermistor elastomer sensor, encompassing the excellent thermal resolution and deformability along with obvious resistance signal to accommodate dynamic detection states is of great importance in thermistor electronic applications. Here the authors proposed a nacre-mimetic laminated strategy to realize a versatile MXene-based thermistor elastomer sensor. The authors claim that such sensor delivers competitive advantages of superior thermosensitivity ($-1.32\% \text{ } ^\circ\text{C}^{-1}$), outstanding temperature resolution ($\sim 0.3 \text{ } ^\circ\text{C}$), and unparalleled mechanical durability (20000 folding fatigue cycles), together with considerable improvement in strain-tolerant thermosensation over commercial thermocouple in exercise scenario. Overall, the topic presented in this work is interesting, however the fundamental contribution is low since the explanation about the mechanism of the superior properties from structural aspect is not sound. This study in itself did not represent the sufficient novelty of the work, and the comparison with other previous strategies is not sufficient to demonstrate the advantages of the current strategy. In addition, the following comments should be addressed before further considering the paper.

Response: We sincerely appreciate the reviewer for the critical evaluation of our manuscript and the recognition of interesting point. We are fully aware of the importance of the reviewer comments regarding the readiness of our manuscript for publication, especially the elaboration about the mechanism from structural aspect. This is closely related to the reviewer's point #3 about the heterostructure and in-plane stress dissipation, and we address it by supplying the new experimental results. To relief the reviewer concern of novelty, we sought to briefly clarify as follows:

Response to the novelty of this study:

(1) The advancement of our work in comparison with previous structure in terms of simplicity preparation method, outstanding performances integration, and general structural paradigm.

To date, extensive efforts have been devoted to exploring high resolution thermistor elastomer sensor (TES) with intrinsic disturbance immunity via engineering structural strategies including kirigami, isolated island, serpentine, and wrinkle structure owing to their unique arrangements that facilitate the stress dissipation and mitigate the deformable magnitude of elastomeric substrate or active layer. Despite the validity of above strategies have been verified, it is still challenging for the realistic implementation because of the following underlying concerns:

1) The complicated instruments and time-consuming operations, such as multistep photolithography, vacuum deposition process, encapsulation, and pattern growth, greatly hinder the widespread deployment of TES.

2) The most available TES that based on above strategies typically possess either superior thermosensitivity or high mechanical resilience, but not both. The comparison of the advantages of interlocking laminated architecture in this work with previous structural strategies has been demonstrated in **Fig. 51 and Supplementary Table 3**. It

is particularly to point out that the kirigami structures are prone to structural damage under large deformation (**Fig. R4**). The crack propagation and breakage, puncture by sharp objects, and accidental scratches that invariably occur during long-term services, would cause the structural vulnerability and compromise the service reliability.

Fig. R4. Visual comparison of stable interlocking laminated architecture with the vulnerable kirigami structure.

3) Strategies that rely on tedious structural designs and special material compositions do not share customizable and universal characteristic for fabricating the thermistor electronics towards wide applicability and commercialization.

In our work, we overcome the undesirable strain interference and solve the structural vulnerability of TES through a facile layer-by-layer assembly and interlocking technique that engineers nacre-mimetic laminated architecture paradigm for high-fidelity temperature indication and embraces a variety of static and dynamic scenarios with excellent strain-unperturbed thermal response ($-1.32\% \text{ } ^\circ\text{C}^{-1}$), and unique mechanical durability (20000 cycles). Additionally, we substantiate some additional results to demonstrate the benefits of interlocking laminated architecture for performances optimization including:

- Structural evolution of PVA and PTF/MXene/Fe composites after the large deformation (verified the efficient interfacial bridging).
- Strain/pressure sensitivity comparison of TES with interlocking laminated architecture and TES with homogeneous structure.
- Puncture resistance test (confirmed the interlocking for preventing interlayer slippage and crack propagation).
- Correlation of sensing layer number with strain sensitivity and mechanical performances (revealed structural optimization).

Overall, combined with the above new experimental progress we firmly believe that this work represents a major advancement in thermistor electronics in terms of facile and scalable fabrication processes, performances collection, and architecture paradigm for broad selection of thermosensation building blocks.

(2) The demonstration of universal nacre-mimetic architecture combined with efficient interlocking for robustness and strain-unperturbed thermistor electronics.

Another novelty of this study is the efficient interlocking of nacre-mimetic architecture. Nacre is a well-defined natural model for mechanical study, and the staggered arrangement of discontinuous micro-platelets that feature pull out and slippage for toughening attracts considerable attention. While the current reports have verified that the synergetic toughening in the nacre-like hierarchical assemblies can facilitate the energy dissipation against the impacts impinging, and further modulate the crack propagation or piezoresistive mediated sensitivity, but it still remains elusive and needs to be explored. Our work elucidates the underlying mechanisms of both heterogeneous laminated structure characteristics and toughening interface for strain insensitivity and mechanical resilience, thus will drive the innovation of universal interlayer coupling methods for high performances nacre-mimetic materials.

1. *The nacre-mimetic laminated strategy is known to be a template to amplify the toughness. The reported thermosensation (-1.32 % °C⁻¹), superior temperature resolution (0.3 °C), and ultrafast responsive time (7 s) seem to mainly depend on the impressive thermosensation mechanism relied on the thermally activated tunnelling current passed through MXene nanosheet junctions as claimed by the authors. The authors need to provide more evidences to clarify the enhanced thermosensation mainly determined by the thermosensitive MXene nanosheets or the nacre-mimetic laminated structure.*

Response: We appreciate this valuable suggestion. The thermal response of TES mainly attributes to the addition of thermosensation components (MXene nanosheets), which is verified via electrochemical impedance spectroscopy (EIS) and I-V curves, and the similar results have been widely witnessed in many previous reports (just name a few of them):

<https://dx.doi.org/10.1021/acs.jpcclett.0c02886>

<https://doi.org/10.1016/j.cej.2021.131699>

<https://doi.org/10.1002/adma.202008308>

<https://doi.org/10.1038/s41528-022-00140-4>

<https://dx.doi.org/10.1021/acs.nanolett.0c02519>

The electrochemical properties of PVA, PTF, PTF/MXene, and PTF/MXene/Fe composites are characterized by EIS where the intercepts of EIS curves with x axis can be considered as the impedance. It is observed that the PTF/MXene/Fe composites possess the lowest impedance (**Supplementary Fig. 13a**), evidencing that the MXene nanosheets play the key role in electrochemical performances. **Supplementary Fig. 13b and c** further present the I-V curves and current values at 1 V of PVA, PTF, PTF/MXene, and PTF/MXene/Fe composites at 20 and 40 °C, where values of PVA and PTF composites remain virtually constant. In contrast, the current values of PTF/MXene, and PTF/MXene/Fe composites dramatically increase (from 2 to 4.43, and 2.4 to 5.57, respectively), which is consistent with the increasing tendency of temperature. These findings collectively suggest that the thermosensation mainly is ascribed to presence of MXene nanosheets.

Updated Supplementary Fig. 13 in Supplementary Information:

Supplementary Fig. 13. **a** The electrochemical impedance spectroscopy (EIS) of PVA, PTF, PTF/MXene, and PTF/MXene/Fe composites, respectively, and **b** measured I–V curves of these composites at 20 °C and 40 °C. **c** Comparison of current variations at 1 V from 20 to 40 °C for PVA, PTF, PTF/MXene, and PTF/MXene/Fe composites, respectively.

Additionally, our work illustrates the role of compact interlocking laminated architecture on thermal responsiveness as the reviewer mentioned. We further fabricate and examine a type of reference sensor, the homogenous elastomer sensor, which keeps the same TOCNF, MXene nanosheets, and Fe contents, only without layer-by-layer assembly and interlayer spraying processes. As a result, this homogeneous elastomer sensor shows resistance variation (~ 34.2) associated with relatively large response time (~ 20 s) from 20 to 40 °C (**Supplementary Fig. 14**), while the TES in conjunction with LBL assembly that sharing MXene nanosheets as thermal paths possess obvious variation value about 39.4 with short response time (~ 13.5 s). This phenomenon reflects that the tight packing of MXene nanosheets via the LBL process may facilitate the construction of thermally conductive pathways and reduces the energy barriers for electron hopping.

Updated Supplementary Fig. 14 in Supplementary Information:

Supplementary Fig. 14. Comparison of temperature response for TES with homogeneous elastomer structure and interlocking laminated architecture from 20 to 40 °C.

2. The authors claim that restraining stress concentration in the proposed biomimetic assemblies can enhance strain insensitivity and overcome the compromising between deformability and thermosensation. Does the nacre-mimetic laminated structure is optimal? Is there any optimal size ratio of the nacreous laminated structure?

Response: We thank the reviewer for the comments. As we proposed, the strain insensitivity is mainly due to the heterogeneous laminated structure (related with layer number) and interfacial bridging (related with TOCNF and metal ion contents). Particularly, the mass ratio of FeCl₂ to MXene nanosheets is given at 3:8 for preventing the excessive destruction of electrostatic repulsive force between the MXene nanosheets (<https://doi.org/10.1002/adma.201902432>). In the revised manuscript, we have quantified the structural optimization about layer number and TOCNF content from both mechanical toughening and strain insensitivity performances.

From mechanical resilience performances:

The sensing layer optimization demonstrates the dramatic improvement in robustness by readily altering the layer number (the corresponding thickness from 0.12 to 0.99 mm) that attains the maximum tensile strain and stress of 1694.8 % and 5.76 MPa, respectively (**Supplementary Fig. 10b**). Additionally, the uniaxial tensile tests corroborates that the proportion of TOCNF (from 0.25 to 1.25 wt %) is crucial to further optimize the mechanical properties across a wide range including tensile stress from 3.38 to 6.68 MPa and Young's modulus from 0.11 to 0.39 MPa, respectively. (**Supplementary Fig. 10c**) Simultaneously, the pronounced increasing of elasticity with the addition of Fe(II) is demonstrated in all counterpart samples, convincing the desirable reinforcement effect of the coordination bonds (**Supplementary Fig. 10d**). In view of the skin-like Young's modulus (0.35 ± 0.032 MPa) and a relatively high toughness (50.62 ± 1.53 MJ m⁻³), the TOCNF content of 1 wt% and is optimal for the subsequent context.

Updated Supplementary Fig. 10 in Supplementary Information:

Supplementary Fig. 10 Mechanical optimization. **b** The laminated layer number (1, 4, 8, and 16 layers, respectively) dependent tensile stress–strain curves of pristine PVA with alternating laminated architecture. **c** The typical tensile stress–strain curves of PTF/MXene and PTF/MXene/Fe composites with different TOCNF contents (0.25, 0.5, 0.75, 1, and 1.25 wt%, respectively), and **d** the corresponding TOCNF content dependent Young's modulus and tensile stress, error bars were defined as S.D. ($n = 3$ independent samples).

From strain insensitivity performances:

From a more quantitative analysis, the strain-unperturbed performances can be optimized by tailoring the structural parameters like the layer number (Fig. 5e). Typically, the TES shows diminishing strain sensitivity from 0.89 to 0.36 with the increasing of the layer number (1-16, thickness from 0.12 to 0.99 mm), while the tensile stress at 100% deformation possesses the contrary increasing tendency from 426 to 876 kPa. These results are consistent with the finite element simulation (Fig. 4e), meaning that the robust multiple layers as building blocks not only act as the electron transfer barrier for inhibiting conductive paths formation, but also contribute to the synergistic toughening in conjunction with interfacial bridging for restraining structural variations, thereby realizing the strain-unperturbed mechanism. Considering the excessive layers (i.e. thickness increasing) inevitably compromises the mechanical harmony with skin because of the excessive rigidity, the eight-layer of alternating laminated architecture that shows obvious strain insensitivity is chosen as the optimized configuration.

Updated Supplementary Fig. 10 in Supplementary Information:

Supplementary Fig. 30. The relative resistance variations and strain sensitivity of TES with different sensing layer number (1, 4, 8, and 16, respectively) under 100 % deformation.

Updated Fig. 5e in Manuscript:

Fig. 5 e The correlation of sensing layers number of TES with strain sensitivity and stress at 100 % deformation, along with the thickness increasing.

3. How do the authors model the mechanical response of TES with alternating laminated architecture with undulating shape under normal stress by using FEM simulation? How do the authors deal with the interface between the laminates? What mechanism can be derived by analyzing the stress field in the laminated structure?

Response: We thank the reviewer for stating their confusion on FEM simulation results. Since FEM simulation is strong evidence for revealing the manner of the strain insensitivity, we make further supplement and discussion to better elucidate the advantages of proposed architecture design and interlocking strategy.

(1) Model construction and simulation conditions setting.

We performed finite element simulation using ABAQUS/Standard for the constructed model. The sensing layer of PTF/MXene/Fe composites with alternating

laminated architecture were modelled that satisfied with Lagrangian formulation with predetermined parameters including density (ρ) = 1.3 g/cm³, Young's modulus (E) = 0.35 MPa, Poisson ratio (ν) = 0.35 (**Supplementary Note 2**). Considering the topology interlocking of the multiple layer architecture, all simulation conditions were applied to the whole model and assumed as a linear elastic material without plastic behavior, referenced to:

<https://doi.org/10.1038/s41563-022-01292-4>

<https://doi.org/10.1021/acs.nanolett.1c03241>

<https://doi.org/10.1002/aenm.202102993>

<https://doi.org/10.1021/acsnano.1c01606>

<https://doi.org/10.1002/adma.202106212>

Moreover, we used a free mesh with triangular elements for all the regions. We seeded at least five nodes on the smallest features of the geometry. In the topological interlocking area between adjacent layer, the mesh was locally refined (100 nodes) such that the relation between the contact areas and applied force can be well captured. The element type is a second-order triangular (6-node) plane-strain modified hybrid element in ABAQUS. The computational process of ABAQUS simulation is presented in Fig R5.

- Artificial strain energy: ALLAE for Whole Model
- Contact constraint discontinuity work: ALLCCDW for Whole Model
- Contact constraint elastic energy: ALLCCE for Whole Model
- Contact constraint elastic normal energy: ALLCCEN for Whole Model
- Contact constraint elastic tangential energy: ALLCCET for Whole Model
- Contact constraint stabilization dissipation: ALLCCSD for Whole Model
- Contact constraint stabilization normal dissipation: ALLCCSDN for Whole Model
- Contact constraint stabilization tangential dissipation: ALLCCSDT for Whole Model
- Creep dissipation energy: ALLCD for Whole Model
- Damage dissipation energy: ALLDMD for Whole Model
- Dynamic time integration energy: ALLDTI for Whole Model
- Electrostatic energy: ALLEE for Whole Model
- Energy lost to quiet boundaries: ALLQB for Whole Model
- External work: ALLWK for Whole Model
- Frictional dissipation: ALLFD for Whole Model
- Internal energy: ALLIE for Whole Model
- Joule heat dissipation: ALLJD for Whole Model
- Kinetic energy: ALLKE for Whole Model
- Loss of kinetic energy at impact: ALLKL for Whole Model
- Plastic dissipation: ALLPD for Whole Model
- Static dissipation (stabilization): ALLSD for Whole Model
- Strain energy: ALLSE for Whole Model
- Total energy of the output set: ETOTAL for Whole Model
- Viscous dissipation: ALLVD for Whole Model

Fig. R5. The computational process of ABAQUS simulation.

The predefined vertical stress field (Z axis) was applied as an exemplification to mimic the most frequently encountered force of skin in daily life (squeeze, touch, and bump). The boundary conditions were set at the bottom of the architecture model (**Fig. R6** of screenshot for ABAQUS simulation), and gradually increased smooth downward velocity of the model with the final displacement on Z direction of 200 μm within 0.2 s and an average velocity of 1 mm/s.

Fig. R6. Screenshot of alternating laminated architecture with obvious boundary settings in ABAQUS simulation.

(2) Interfacial bridging between the adjacent layer.

In order to dynamically manifest the stress distribution in nacre-mimetic architecture during distortion, we developed a 3D finite element model using CINEMA 4D R20 software for the nacre-like structure that duplicated the “brick-and-mortar” arrangement from mollusk shells to achieve the finite element simulation (ABAQUS). With regard to the interface of adjacent layer, we designed the layer volume network combined with random volume growth strategy to mimic the random cross-linking between adjacent layers (**Fig. R7 and R8**), thus the adjacent layer possessed random and discrete interface gap as default option in ABAQUS simulation (**Fig. R9**).

Fig. R7. Volume network design in CINEMA 4D R20.

Fig. R8. Models and random interfacial bridging in CINEMA 4D R20.

Fig. R9. Random and discrete interface gap in ABASQUS simulation.

(3) In-plane stress dissipation mechanism.

To achieve the strain insensitivity, we focus on the synergistic effects between the interfacial bridging and the multilayer polymer barrier in nacre-like structures, where the interlocking can restrain linear deformation over wearable application and is the key to the strain-unperturbed temperature monitoring.

As shown in **Fig. R10** of the Z axis deformation process in ABASQUS simulation, the main borne regions of compression loading upon topological interlocking, stress distributions at the interface gap, and in-plane deflection and branching can be observed in sequence in the nacre-mimetic architecture model. Accompanying with external loading, the stress dispersion following tortuous paths (dynamic crosslinking) is distinctly intensified throughout enlarged contact regions of adjacent layers (green color deepened and overflowed).

Fig. R10. Stress distribution nephograms of interfacial bridging in alternating laminated architecture during Z axis deformation process (see dynamic evolution process in **Supplementary Movie S1**).

In order to intuitively clarify the mechanism of in-plane stress dissipation, the alternating laminated architecture under different external loadings are displayed in out-plane view, cross-section view, and in-plane view, respectively (**Supplementary Fig. 24**). From the in-plane view, the stress dispersion path is tortuous along the interface between interlocked adjacent layers (bright color regions) for preventing the crack propagation that accompanied with the nonlinear deformation (8.92 - 42.5 %) of total architecture.

Supplementary Fig. 24 Finite element simulation results of TES with alternating laminated architecture under normal stress. **a** The out-plane view, **b** cross-section view, and **c** in-plane view of stress diffusion at different deformation ratios. **d** Deformation ratio dependent loading force (left axis) and displacement (right axis) in finite element simulation. **e** Schematic diagram of energy dissipation among adjacent layers via multiple hydrogen bonding interactions, coordination bond, and ionic interaction.

On the one hand, the high crystallization of PVA matrix combined with the reinforcement of TOCNF endow the heterostructure unit with intrinsic high strength, enhanced stiffness and toughness, together with multiple layer barrier, restraining the structural variation and conductive path formation. On the other hand, the coordination bonds further reinforce the interfacial interactions between adjacent layer, which can subsequently absorb larger amounts of mechanical energy and facilitate the stress dispersion to hinder crack initiation upon external loading.

In summary, the structural variations accompanying with conductive paths formation demonstrate monotonical changes of resistance signal, while the interlocking laminated architecture combined with the efficient in-plane stress dissipation achieve the deformation suppression and the electron transfer inhibition, thus demonstrates as the deepened and overflowing color at interface of adjacent layer in finite element simulation for strain insensitivity.

4. The authors claim that significantly alleviated the strain interference is due to the biomimetic laminated strategy combining with the in-plane stress dissipation and nacre-mimetic hierarchical architecture. However, as is shown in Figure 5 significant relative resistance change due to the applied pressure or strain is still observed. The strain sensitivity did not vanishes. The strain signal decoupling mechanism is not sufficiently clarified in the present manuscript.

Response: We thank the reviewer for stating this point. In order to satisfy with wearable characteristic, the stretchable thermistor elastomer materials inevitably face the deformation mediated electrical signals fluctuation, which can only be suppressed to a relative low level for avoiding strain interference. Although the pressure or strain mediated relative resistance variations can still be observed in our work, the waveform and amplitude are difficult to disturb the temperature mediated electrical signals (**Fig. 5j**), and the strain/pressure sensitivity is much lower than the level of most previous works in literature (**Table R2** in point #8 for reviewer 1).

To quantitatively decipher the strain insensitivity of TES, both pressure sensitivity (S) and gauge factor (GF) are calculated, which defined as the ratio of relative resistance changes ($\Delta R/R_0$) to the applied pressure (P) or strain (ε), respectively. A type of reference sensor, defined as TES with homogeneous structure, only in absence of layer-by-layer assembly and interlayer spraying processes is fabricated for comparison. **Fig. 5a** depicts the $\Delta R/R_0$ as a function of the applied pressure levels, and TES with homogeneous structure shows a two-stage linear response (1.53 kPa^{-1} from 0 to 12 kPa and 0.088 kPa^{-1} from 12 to 100 kPa), while TES with interlocking laminated architecture presents an extremely weakened piezoresistive effect (0.66 and 0.075 kPa^{-1} in both 0-20 kPa and 20-100 kPa regions). The pressure sensitivity is reduced about 2.3 times and the sensing range is extended to 0-20 kPa in the first piezoresistive response interval.

Except that, compared to the TES with homogeneous structure, the resistance variations of TES with interlocking laminated architecture decrease about 17.92 % over the entire skin stretching range (0–100%), together with limited detection sensitivity (GF ~ 0.467) that far below the level of most previous flexible electronics. It is undeniable that the high crystallinity PVA polymer sensing layer leads to the intrinsically weak conductive paths, while the above solid results verify that the proposed alternating laminated architecture in conjunction with interfacial bridging successfully contribute to the strain insensitivity performance.

Updated Fig. 5a and d in Manuscript:

Fig. 5 a The weak piezoresistive effect in the applied pressure region (0-100 kPa), and **d** unobvious gauge factor (GF) in 100% strain range for TES with interlocking laminated architecture in contrast to the TES without structural designs.

We further supplement the discussion about the differences among the homogeneous structure, nacre-mimetic architecture, and interlocking laminated architecture to understand the unique strain insensitivity that achieved via compact heterostructure and efficient interlocking.

1) Homogeneous structure

The thermistor elastomer sensors with homogeneous structure achieve the energy dissipation through the polymer chains entanglement and dissociation, as well as the dynamic bonding, while the stress dispersion in the matrix along with obvious structural variation and conductive pathway construction lead to strain interference for temperature response.

2) Nacre-mimetic architecture

A representative strategy enabled strain insensitivity relies on the structural heterogeneities, like nacre mimetic architecture, can restrain the conductive paths formation through adjacent layer barrier and achieves the mechanical toughening via the layer-by-layer energy dissipation manner. However, the lack of efficient interfacial bridging leads to the interface failure and crack propagation among multiple layers, which in turn affects the toughness and deformation resistance of nanomaterials.

3) Interlocking laminated architecture

In our work, the interlocking nacre-mimetic architecture achieves the synergistic toughening and strain insensitivity. The efficient interlocking in conjunction with the adjacent layer barrier affects the stress distribution and restrains the construction of conductive path under external loading, thus the prepared thermistor elastomer sensor possesses excellent thermosensation without strain induced signal distortion.

Updated Supplementary Fig. 26 in Supplementary Information:

Supplementary Fig. 26. The comparison of stress dispersion under external loading in the homogeneous structure, nacre-mimetic architecture, and interlocking laminated architecture.

5. In addition, how do the authors believe that it is the biomimetic laminated strategy combining with the in-plane stress dissipation and nacre-mimetic hierarchical architecture not other factor that delivers competitive advantages of superior properties?

Response: we sincerely appreciate the reviewer's feedback, this question is similar to the point #8 from reviewer 1, and it is adequately validated and theoretically evidenced through supplementary data including puncture resistance test for interfacial bridging, structural evolution observation after deformation, quantitative structural optimization, deformation insensitivity performances comparison, and finite element simulation discussion.

A brief explanation of proposed strategy for strain insensitivity:

In this work, we demonstrate that the interfacial bridging (mainly coordination bonds) of compact laminated architecture enables the obtained thermistor elastomer remarkable interlayer coupling capability and high stress transfer efficiency in tandem with multi-layer polymer barrier, collectively lead to the mechanical toughening and strain insensitivity. As evidenced in **Fig. 4d**, a pronounced hysteresis loop with exceptional recovery efficiency (~73.57 %) of PTF/MXene/Fe composites is noted from the successive loading-unloading curves in comparison with that of PVA even after 100 fatigue cycles at strain of 500 %. The robust interface of PTF/MXene/Fe in SEM images in contrast to the delamination and slippage between adjacent layers of PVA further infers the efficient interfacial bridging for stress dissipation. In addition, the energy dispersive spectrometer (EDS) mapping in **Supplementary Fig. 29** provides direct clues that the Fe element as the interfacial bridging agents predominantly exists in the adjacent layers, clarifying the key role of coordination bonds in facilitating the stress dispersion.

Updated Fig 4.d in Manuscript:

Fig.4 d Cyclic loading–unloading tensile curves (100 cycles, 500 % strain) for PTF/MXene/Fe and PVA composites, the inset showing the cross-section SEM images after 100 stretching cycles.

Updated Supplementary Fig. 28 in Supplementary Information:

Supplementary Fig. 28 The EDS mapping images for uniform distribution elements of C, O, and Ti, and the detailed spatial distribution of Fe as interfacial bridging agents.

In addition, we find the performance of interference-free temperature monitoring can be tailored by the structural variables of interlocking laminated architecture, such as interfacial bridging and layer number. Indeed, the TES with interlocking laminated architecture presents a more weakened piezoresistive effect (0.66 and 0.075 kPa^{-1} in both $0\text{-}20 \text{ kPa}$ of low-pressure range and $20\text{-}100 \text{ kPa}$ of high-pressure range, respectively), which is reduced about 2.3 times in comparison with homogeneous TES (**Fig. 5a**). In the piezoresistive effect, pressure sensitivity is strongly correlated with the quantity of conductive pathways that form in the piezoresistive material, while the tough interface not only ensures the stable layer barrier but also avoids the contact area variation induced conductive path formation.

To further justify this point, the puncture resistance test is conducted because this out-of-plane loading can partly reflect the microscale interlayered interaction of the nanocomposites that is prepared via layer-by-layer technique. As demonstrated in **Fig. 5c**, the PTF/MXene/Fe composites could withstand a puncture force as high as 13.42 N, which is superior to PVA, PTF, and PTF/MXene composites (3.9, 6.7, and 7.6 N, respectively). It can be speculated that the poor interlayered stress transfer would cause to high shear stress and unexpected delamination that enlarge the extent of damage region, thus cause obvious structural deformation. Besides, the linear resistance variation of TES with interlocking laminated architecture over the entire skin stretching range (0–100%) is decreased by 17.92 %, together with a limited detection sensitivity (GF ~0.467) in contrast to the homogeneous TES (**Fig. 5d**). Furthermore, the TES shows diminishing strain sensitivity from 0.89 to 0.43 with the increasing of the layer number (from 1 to 16) in the deformation range of 1-100 % (**Fig. 5e**), while the tensile stress at 100% deformation increases from 426 to 876 kPa, inferring the multiple adjacent layer barrier and interface crosslinking synergistically contribute to the toughening mechanism and suppress the crack propagation along the deformation direction for the purpose of strain insensitivity. Moreover, the FEM analysis is conducted to elucidate the strain-unperturbed mechanism by exploring the stress distributions in the architecture upon external loading (**Supplementary Fig. 24**). When the architecture is stressed, the topological interlocking promotes the stress transfer that demonstrate as numerous in-plane stress diffusion (noted as arrows and enlarged color regions) following tortuous paths (dynamic crosslinking), inferring the structural variation mediated electronic output has been suppressed.

Overall, systematic structural evolution and mechanical tests clearly demonstrate that the heterogeneous architecture with efficient interlocking enables the resultant thermistor elastomer remarkable interlayer coupling capability, high stress transfer efficiency, tortuous crack propagation, leading to synergetic toughening mechanism and in-plane stress dissipation for strain-unperturbed thermosensation.

Updated Fig 5 in Manuscript:

Fig.5 a The weak piezoresistive effect in the applied pressure region (0-100 kPa) for TES with/without interlocking laminated architecture. **b** Puncture resistance test (needle diameter = 900 μm , loading speed = 50 mm/min) of PVA, PTF, PTF/MXene, and PTF/MXene/Fe composites, respectively. **c** Unobvious gauge factor (GF) in 100 % strain range for TES with interlocking laminated architecture in contrast to the TES with homogeneous structure. **d** The correlation of sensing layer number of TES with strain sensitivity and stress at 100 % deformation, along with the thickness increasing.

6. The colorbar shown in Fig.1(d) is unclear.

Response: Thanks for pointing it out. We improve the image resolution of Fig. 1(d) for better clarity.

Updated Fig. 1d in Manuscript:

7. The phrase of “the corresponding in-plane stress diffusion” shown in Fig. 4(d) is unclear.

Response: Thanks for the advice, and we revise the description of **Fig. 4e**.

Updated Fig. 4e in Manuscript:

(e) Finite element simulation of stress distribution within alternating laminated architecture under the vertical uniaxial press of different deformation ratios (9, 14.5, 25.5, 30.6, and 42.5 %, respectively).

REVIEWER COMMENTS

Reviewer #2 (Remarks to the Author):

The authors have well addressed my comments. I am satisfied with their response. I recommend publication of the paper without further review.

Reviewer #3 (Remarks to the Author):

By adding additional experimental results and discussion to the revised manuscript, the authors have elaborated the novelty of the work. The comments have been carefully addressed. Therefore we have no further comments before being considered for publication.

Point-to-Point Responses to Referees' Comments

Reviewer #2 (Remarks to the Author):

The authors have well addressed my comments. I am satisfied with their response. I recommend publication of the paper without further review.

Response: Thank you for your positive comments.

Reviewer #3 (Remarks to the Author):

By adding additional experimental results and discussion to the revised manuscript, the authors have elaborated the novelty of the work. The comments have been carefully addressed. Therefore, we have no further comments before being considered for publication.

Response: Thank you for your positive comments.